# Affine Steerable Equivariant Layer for Canonicalization of Neural Networks

**Yikang Li**[1], **Yeqing Qiu**[2,3], **Yuxuan Chen**[4], **Zhouchen Lin**[1,5,6*]
[1]State Key Lab of General AI, School of Intelligence Science and Technology, Peking University
[2]The Chinese University of Hong Kong, Shenzhen  [3]Shenzhen Research Institute of Big Data
[4]Khoury College of Computer Sciences, Northeastern University
[5]Institute for Artificial Intelligence, Peking University
[6]Pazhou Laboratory (Huangpu), Guangzhou, Guangdong, China

## Abstract

In the field of equivariant networks, achieving affine equivariance, particularly for general group representations, has long been a challenge. In this paper, we propose the steerable *EquivarLayer*, a generalization of InvarLayer (Li et al., 2024), by building on the concept of *equivariants* beyond invariants. The steerable EquivarLayer supports affine equivariance with arbitrary input and output representations, marking the first model to incorporate steerability into networks for the affine group. To integrate it with canonicalization, a promising approach for making pre-trained models equivariant, we introduce a novel *Det-Pooling* module, expanding the applicability of EquivarLayer and the range of groups suitable for canonicalization. We conduct experiments on image classification tasks involving group transformations to validate the steerable EquivarLayer in the role of a canonicalization function, demonstrating its effectiveness over data augmentation.

## 1 Introduction

Convolutional neural networks (CNNs) have achieved remarkable success across domains (He et al., 2016; Chen et al., 2017; Ren et al., 2016; Raissi et al., 2019), largely due to their property of translation equivariance. To incorporate broader symmetries into networks, Cohen & Welling (2016a) introduce Group Equivariant CNNs (G-CNNs), which conduct convolutions on groups, preserving rotation symmetry. Steerable CNNs (Cohen & Welling, 2016b; Weiler & Cesa, 2019) generalize G-CNNs, allowing for arbitrary input and output representation types by viewing features as fields. Further works extend equivariance to subgroups of Euclidean groups across diverse domains (Esteves et al., 2019; Weiler et al., 2018; Wang et al., 2022; Cohen et al., 2018; 2019). Recently, canonicalization has emerged as a novel and promising method for enforcing equivariance or invariance (Kaba et al., 2023). It maps inputs to a canonical form, based on the group element output by a canonicalization function, before passing them through a non-equivariant network. This allows models to exhibit equivariance without architectural changes or retraining, thus enabling equivariant adaptation of pre-trained models (Mondal et al., 2023). Currently, this method has been primarily applied to Euclidean groups, as the canonicalization function is typically constructed using equivariant networks with specific non-trivial group representations. Extending equivariant networks with steerability to more general groups can significantly broaden the applicability of canonicalization.

Achieving equivariance for more general groups, particularly the affine group, remains a significant challenge. MacDonald et al. (2022) enable group convolution on the affine group by converting the integral over the group into one over the Lie algebra, but it still requires group sampling, leading to exponential memory growth. Recent work by Mironenco & Forré (2024) improves sampling efficiency by decomposing large groups, yet limited to regular representations. Li et al. (2024) propose InvarLayer, based on differential invariants, achieving affine equivariance in networks without discretizing or sampling the group for the first time. However, InvarLayer is inherently limited to the trivial representation for input and output. It is still a challenge to design a layer that can handle affine equivariance for arbitrary input and output representation types. A more detailed literature review can be found in Appendix A.

---

[*]Corresponding author.

In this paper, we extend the InvarLayer (Li et al., 2024) framework to develop a more general equivariant layer, called the steerable *EquivarLayer*. This extension is analogous to the generalization from G-CNNs (Cohen & Welling, 2016a) to steerable CNNs (Cohen & Welling, 2016b; Weiler & Cesa, 2019), and from PDO-eConvs (Shen et al., 2020) to steerable PDOs (Jenner & Weiler, 2021). While invariants, the core concept in InvarLayer, only generate equivariant operators that produce features tied to the trivial group representation, we employ a generalized concept beyond invariants, referred to as *equivariants*. Unlike invariants, which remain unchanged under group transformations, equivariants transform according to given group representations, thus yielding more general equivariant operators that can map between different feature types. To construct equivariants, we introduce the notion of an *equivariant matrix*, which, when multiplied with invariants, generates equivariants. With existing methods for computing invariants (Olver, 2015; Wang et al., 2013; Li et al., 2024), our focus turns to building equivariant matrices. We apply the tool of moving frames to construct them, similar to the process for constructing equivariant maps in (Peter J. Olver, 2024). Additionally, inspired by the construction of SupNorm normalized differential invariants (Li et al., 2024), we introduce a novel normalization technique to enhance the numerical stability of equivariant matrices. Based on equivariants, we design the steerable EquivarLayer, enabling flexible specification of input and output features corresponding to arbitrary group representations. The steerable EquivarLayer supports equivariance to the affine group and its continuous subgroups, marking the first instance of affine steerable equivariant models that allow for flexible input and output representation types, filling a gap in the field of equivariant networks.

Building on this, we apply the steerable EquivarLayer as the canonicalization function within the canonicalization framework. Specifically, the steerable EquivarLayer outputs a matrix field that satisfies the equivariance property required for the canonicalization function. We then introduce a novel module, *Det-Pooling*, to select the appropriate output matrix, enabling the model to handle matrix groups such as $\mathrm{GL}^+(2)$. This approach significantly enhances the flexibility and scope of canonicalization, extending it beyond the traditionally used rotation groups to support generalized group equivariance across a broader range of applications.

We summarize our main contributions as follows:

- Extending the InvarLayer (Li et al., 2024) framework, we design the steerable EquivarLayer based on equivariants, allowing arbitrary specification of input and output representations while supporting the affine group and its continuous subgroups. This is the first time steerability has been introduced to networks for the affine group.
- We propose a new normalization technique that transforms relative equivariants into absolute equivariants, facilitating the construction of simpler equivariant matrices and enabling the generation of more numerically stable equivariants.
- We introduce a novel module, Det-Pooling, which allows the application of the steerable EquivarLayer to canonicalization. This extends the range of groups applicable to canonicalization, making it possible to handle more complex matrix groups.
- We demonstrate the effectiveness of our approach through image classification experiments involving three non-Euclidean groups, including $\mathrm{GL}^+(2)$, the rotation-scale group, and the scale group, showing better performance compared to data augmentation. [1]

## 2 Steerable EquivarLayer

In this section, we introduce the steerable EquivarLayer, extending the InvarLayer framework to support affine equivariance with arbitrary input and output representations.

### 2.1 Basic concepts and notations

In our framework, we treat the features of each network layer as a smooth vector function on a continuous domain. We specifically focus on the most common case of the 2D plane, $\mathbf{u} : \mathbb{R}^2 \to \mathbb{R}^c$, where each point $\mathbf{x}$ is associated with a vector $\mathbf{u}(\mathbf{x})$, referred to as a fiber, and $c$ is the number of channels. We consider a transformation group $G$ that acts on the input space $\mathbb{R}^2$, specifying the symmetry we aim to maintain as equivariance across the layers. Moreover, the features of each layer are associated with a group representation $\boldsymbol{\rho} : G \to \mathrm{GL}(c)$, which characterizes the *type* of the layer and dictates how the $c$ channels of each feature vector $\mathbf{u}(\mathbf{x})$ transform under group actions.

---

[1]The code is available at https://github.com/Liyk127/EquivarLayer.

This property, known as *steerability*, indicates that the layers respond to transformations in a manner consistent with their respective types.

Next, we formally introduce the notions and definitions related to group actions and equivariance. Given a group $G$ and a function space $\mathcal{F} = \{\mathbf{u} \mid \mathbf{u} : \mathbb{R}^2 \to \mathbb{R}^c\}$ with type $\boldsymbol{\rho}$, we denote the group action on the function space corresponding to the group representation $\boldsymbol{\rho}$ as $L_{\boldsymbol{\rho}}(g)$, such that

$$(L_{\boldsymbol{\rho}}(g) \diamond \mathbf{u})(\mathbf{x}) \triangleq \boldsymbol{\rho}(g) \cdot \mathbf{u}(g^{-1} \cdot \mathbf{x}). \tag{1}$$

Here, the group $G$ acts on the base domain $\mathbb{R}^2$ in its standard way. Specifically, we focus on the affine group and its subgroups, where $g \in G$ can be represented as $g = (\mathbf{A}, \mathbf{b})$, with $\mathbf{A} \in \mathbb{R}^{2 \times 2}$ being an invertible matrix and $\mathbf{b} \in \mathbb{R}^2$. We consider the action of $G$ on $\mathbb{R}^2$ as $g \cdot \mathbf{x} = \mathbf{A}\mathbf{x} + \mathbf{b}$. The group action defined in (1) involves not only the transformation of positions but also the transformation of vectors at each point. Unlike the case of scalar fields, each vector is not only moved to a new position but also changes its orientation via the action of $g \in G$. When the type is $\boldsymbol{\rho}_0 = \mathbf{I}$, namely the trivial representation, the vectors retain their orientation, reducing the case to that of scalar fields. With this understanding of group actions, we can now define the concept of equivariance:

**Definition 1** *Let $G$ be a group acting on function spaces $\mathcal{F}$ and $\mathcal{F}'$ with types $\boldsymbol{\rho}$ and $\boldsymbol{\rho}'$, respectively. An operator $\psi : \mathcal{F} \to \mathcal{F}'$ is said to be **equivariant** if*

$$\psi[L_{\boldsymbol{\rho}}(g) \diamond \mathbf{u}] = L_{\boldsymbol{\rho}'}(g) \diamond \psi[\mathbf{u}], \quad \forall \mathbf{u} \in \mathcal{F}, g \in G. \tag{2}$$

An important property of equivariance is its transitivity: the composition of multiple equivariant operators remains equivariant. Consequently, if the mapping between every layer in a network satisfies equivariance with corresponding types, the entire network exhibits steerability.

In (Li et al., 2024), the primary concept is invariants, which also serves as one of the key concepts in this paper. We extend the definition from (Li et al., 2024) as follows:

**Definition 2** *Let $G$ be a group acting on the function space $\mathcal{F} = \{\mathbf{u} \mid \mathbf{u} : \mathbb{R}^2 \to \mathbb{R}^c\}$ with type $\boldsymbol{\rho}$. We define $\mathcal{I} : \mathbb{R}^2 \times \mathcal{F} \to \mathbb{R}$ as an **invariant** if it satisfies*

$$\mathcal{I}(g \cdot \mathbf{x}, L_{\boldsymbol{\rho}}(g) \diamond \mathbf{u}) = \mathcal{I}(\mathbf{x}, \mathbf{u}), \quad \forall \mathbf{x} \in \mathbb{R}^2, \mathbf{u} \in \mathcal{F}, g \in G. \tag{3}$$

*We define $\boldsymbol{\mathcal{I}} \triangleq (\mathcal{I}_1, ..., \mathcal{I}_k)^\top$ as a $k$-dimensional invariant if $\mathcal{I}_1, \ldots, \mathcal{I}_k$ are all invariants of $G$.*

Compared to the definition in (Li et al., 2024), our formulation generalizes the group action on $\mathcal{F}$, while the fundamental properties of invariants and methods for computing them remain unchanged. Both the classical approach of deriving differential invariants (Fels & Olver, 1999; Olver, 2003; 2015) and the SupNorm normalization method described in (Li et al., 2024) are still applicable.

Invariants and equivariance are closely intertwined. The following proposition reveals how to construct an equivariant operator from invariants:

**Proposition 3** *Let $G$ be a group acting on the function space $\mathcal{F} = \{\mathbf{u} \mid \mathbf{u} : \mathbb{R}^2 \to \mathbb{R}^c\}$ with type $\boldsymbol{\rho}$, and let $\boldsymbol{\mathcal{I}} : \mathbb{R}^2 \times \mathcal{F} \to \mathbb{R}^{c'}$ be a $c'$-dimensional invariant. Given $\mathbf{u} \in \mathcal{F}$, the function $\boldsymbol{\mathcal{I}}(\cdot, \mathbf{u})$ can be viewed as an element in $\mathcal{F}' = \{\mathbf{v} \mid \mathbf{v} : \mathbb{R}^2 \to \mathbb{R}^{c'}\}$. We define $\hat{\boldsymbol{\mathcal{I}}} : \mathcal{F} \to \mathcal{F}'$ as*

$$\hat{\boldsymbol{\mathcal{I}}}[\mathbf{u}] \triangleq \boldsymbol{\mathcal{I}}(\cdot, \mathbf{u}). \tag{4}$$

*Then $\hat{\boldsymbol{\mathcal{I}}}$ is an equivariant operator satisfying*

$$\hat{\boldsymbol{\mathcal{I}}}[L_{\boldsymbol{\rho}}(g) \diamond \mathbf{u}] = L_{\boldsymbol{\rho}_0}(g) \diamond \hat{\boldsymbol{\mathcal{I}}}[\mathbf{u}], \quad \forall \mathbf{u} \in \mathcal{F}, g \in G, \tag{5}$$

*where $\boldsymbol{\rho}_0(g) = \mathbf{I}$ denotes the trivial group representation.*

As shown in Proposition 3, $\hat{\boldsymbol{\mathcal{I}}}$ yields an equivariant operator that maps features of type $\boldsymbol{\rho}$ to features of the trivial type $\boldsymbol{\rho}_0$. However, to construct more general equivariant operators that map features from type $\boldsymbol{\rho}$ to type $\boldsymbol{\rho}'$, invariants alone may not suffice. This is the challenge we aim to address.

## 2.2 EQUIVARIANTS

To achieve steerability across network layers, we need to construct equivariant operators capable of mapping features between any specified input and output types. To this end, we introduce a generalized concept beyond invariants, referred to as *equivariants*. Equivariants maintain the required transformation properties when transitioning between different feature types. The formal definition is provided below:

**Definition 4** *Let $G$ be a group acting on the function space $\mathcal{F} = \{\mathbf{u} \mid \mathbf{u} : \mathbb{R}^2 \to \mathbb{R}^c\}$ with type $\boldsymbol{\rho}$. We define a map $\mathcal{E} : \mathbb{R}^2 \times \mathcal{F} \to \mathbb{R}^{c'}$ as a type-$(\boldsymbol{\rho}, \boldsymbol{\rho}')$ **equivariant** if it satisfies*

$$\mathcal{E}(g \cdot \mathbf{x}, L_{\boldsymbol{\rho}}(g) \diamond \mathbf{u}) = \boldsymbol{\rho}'(g) \cdot \mathcal{E}(\mathbf{x}, \mathbf{u}), \quad \forall \mathbf{x} \in \mathbb{R}^2, \mathbf{u} \in \mathcal{F}, g \in G. \tag{6}$$

It is worth noting that an equivariant can be viewed as a $c'$-dimensional invariant when $\boldsymbol{\rho}' = \boldsymbol{\rho}_0$ is the trivial group representation. Unlike invariants, which produce outputs that always remain unchanged under group transformations, equivariants actively transform according to specified group representations. This flexibility allows us to construct more general equivariant operators capable of mapping features between arbitrary types. The relationship between equivariants and equivariant operators can be formalized as follows:

**Proposition 5** *Let $G$ be a group acting on the function space $\mathcal{F} = \{\mathbf{u} \mid \mathbf{u} : \mathbb{R}^2 \to \mathbb{R}^c\}$ with type $\boldsymbol{\rho}$, and let $\mathcal{E} : \mathbb{R}^2 \times \mathcal{F} \to \mathbb{R}^{c'}$ be a type-$(\boldsymbol{\rho}, \boldsymbol{\rho}')$ equivariant. Given $\mathbf{u} \in \mathcal{F}$, the function $\mathcal{E}(\cdot, \mathbf{u})$ can be viewed as an element in $\mathcal{F}' = \{\mathbf{v} \mid \mathbf{v} : \mathbb{R}^2 \to \mathbb{R}^{c'}\}$. Denote $\hat{\mathcal{E}} : \mathcal{F} \to \mathcal{F}'$ as*

$$\hat{\mathcal{E}}[\mathbf{u}] \triangleq \mathcal{E}(\cdot, \mathbf{u}). \tag{7}$$

*Then $\hat{\mathcal{E}}$ is an equivariant operator that satisfies*

$$\hat{\mathcal{E}}[L_{\boldsymbol{\rho}}(g) \diamond \mathbf{u}] = L_{\boldsymbol{\rho}'}(g) \diamond \hat{\mathcal{E}}[\mathbf{u}], \quad \forall \mathbf{u} \in \mathcal{F}, g \in G. \tag{8}$$

As demonstrated in Proposition 5, the equivariant operator $\hat{\mathcal{E}}$ associated with the equivariant $\mathcal{E}$ facilitates more generalized equivariance, enabling us to specify the group actions on both input and output features. The crucial question, therefore, is how to systematically construct such equivariants.

To address this challenge, we introduce the concept of an *equivariant matrix*, which establishes a framework for constructing equivariants from invariants by creating a relationship between the two. The formal definition is provided below:

**Definition 6** *Let $G$ be a group acting on the function space $\mathcal{F} = \{\mathbf{u} \mid \mathbf{u} : \mathbb{R}^2 \to \mathbb{R}^c\}$ with type $\boldsymbol{\rho}$. We define a map $\boldsymbol{\mathcal{M}} : \mathbb{R}^2 \times \mathcal{F} \to \mathbb{R}^{c' \times c'}$ as a type-$(\boldsymbol{\rho}, \boldsymbol{\rho}')$ **equivariant matrix** if it satisfies*

$$\boldsymbol{\mathcal{M}}(g \cdot \mathbf{x}, L_{\boldsymbol{\rho}}(g) \diamond \mathbf{u}) = \boldsymbol{\rho}'(g) \cdot \boldsymbol{\mathcal{M}}(\mathbf{x}, \mathbf{u}), \quad \forall \mathbf{x} \in \mathbb{R}^2, \mathbf{u} \in \mathcal{F}, g \in G. \tag{9}$$

Using this equivariant matrix, we can construct equivariants by combining it with invariants, as formalized in the following theorem:

**Theorem 7** *Let $G$ be a group acting on the function space $\mathcal{F} = \{\mathbf{u} \mid \mathbf{u} : \mathbb{R}^2 \to \mathbb{R}^c\}$ with type $\boldsymbol{\rho}$. Suppose $\boldsymbol{\mathcal{M}} : \mathbb{R}^2 \times \mathcal{F} \to \mathbb{R}^{c' \times c'}$ is a type-$(\boldsymbol{\rho}, \boldsymbol{\rho}')$ equivariant matrix, and $\boldsymbol{\mathcal{I}} : \mathbb{R}^2 \times \mathcal{F} \to \mathbb{R}^c$ is a $c$-dimensional invariant. Denote $\mathcal{E} : \mathbb{R}^2 \times \mathcal{F} \to \mathbb{R}^{c'}$ as*

$$\mathcal{E}(\mathbf{x}, \mathbf{u}) \triangleq \boldsymbol{\mathcal{M}}(\mathbf{x}, \mathbf{u}) \cdot \boldsymbol{\mathcal{I}}(\mathbf{x}, \mathbf{u}). \tag{10}$$

*Then, $\mathcal{E}$ is a type-$(\boldsymbol{\rho}, \boldsymbol{\rho}')$ equivariant, i.e.,*

$$\mathcal{E}(g \cdot \mathbf{x}, L_{\boldsymbol{\rho}}(g) \diamond \mathbf{u}) = \boldsymbol{\rho}'(g) \cdot \mathcal{E}(\mathbf{x}, \mathbf{u}), \quad \forall \mathbf{x} \in \mathbb{R}^2, \mathbf{u} \in \mathcal{F}, g \in G. \tag{11}$$

There are several well-established methods for constructing invariants, such as deriving differential invariants through the classical moving frame approach (Fels & Olver, 1999; Olver, 2003; 2015) and utilizing the SupNorm normalization technique proposed in (Li et al., 2024). With these methods, we can derive a set of basis invariants and further extend them through arbitrary functional combinations to obtain a rich and comprehensive set of invariants for a given group action. Once a non-trivial equivariant matrix $\boldsymbol{\mathcal{M}}$ is obtained, we can combine it with different invariants $\boldsymbol{\mathcal{I}}$ to construct a diverse set of equivariants.

Next, we tackle the challenge of constructing equivariant matrices. To do this, we will employ the tool of *moving frames*. First, we provide the definition of a moving frame.

**Definition 8** *(Olver, 2015) Let $G$ be an $r$-dimensional Lie group acting on an $m$-dimensional manifold $\mathcal{Z}$. A map $\alpha : \mathcal{Z} \to G$ is called a **moving frame** if it satisfies*

$$\alpha(g \cdot z) = \alpha(z) \cdot g^{-1}, \quad \forall z \in \mathcal{Z}, g \in G. \tag{12}$$

The following theorem provides a method for the practical computation of a moving frame.

**Theorem 9** *(Olver, 2015) Let $G$ act freely and regularly on $\mathcal{Z}$, and let $\mathcal{K}$ be a cross-section. For $z \in \mathcal{Z}$, let $g = \alpha(z)$ be the unique group element mapping $z$ to this cross-section:*

$$g \cdot z = \alpha(z) \cdot z \in \mathcal{K}. \tag{13}$$

*Then $\alpha : \mathcal{Z} \to G$ is a moving frame for the group action.*

The group action is free if, for any $z \in \mathcal{Z}$, the only group element that leaves $z$ unchanged is the identity. The group action is regular if the orbits form a regular foliation of $\mathcal{Z}$. A cross-section is a subset of $\mathcal{Z}$ defined by fixing $r$ coordinates, expressed as $\mathcal{K} = \{z_1 = c_1, ..., z_r = c_r\}$, where $c_1, \ldots, c_r$ are constants and $r = \dim G$. This cross-section intersects each group orbit exactly once and transversely, providing a unique representation for each orbit. For more detailed and rigorous definitions, please refer to (Olver, 2015).

With Theorem 9, we can now apply it in our context. Consider the manifold $\mathcal{Z}$ as the jet space defined by $\mathcal{Z} = \{z \mid z = (\mathbf{x}, \mathbf{u}(\mathbf{x}), \nabla \mathbf{u}(\mathbf{x}), \cdots, \nabla^d \mathbf{u}(\mathbf{x})), \mathbf{x} \in \mathbb{R}^2\}$. The group action on $\mathcal{Z}$ is given by the prolongation of the group action on the function space $\mathcal{F}$. Let $\mathbf{u}_g = L_{\boldsymbol{\rho}}(g) \diamond \mathbf{u}$. Explicitly, the action $z \mapsto g \cdot z$ is defined as follows:

$$(\mathbf{x}, \mathbf{u}(\mathbf{x}), \nabla \mathbf{u}(\mathbf{x}), \cdots, \nabla^d \mathbf{u}(\mathbf{x})) \mapsto (g \cdot \mathbf{x}, \mathbf{u}_g(g \cdot \mathbf{x}), \nabla \mathbf{u}_g(g \cdot \mathbf{x}), \cdots, \nabla^d \mathbf{u}_g(g \cdot \mathbf{x})).$$

With this setup, we can then compute the corresponding moving frame $\alpha$. The following theorem reveals how to use this moving frame to construct an equivariant matrix.

**Theorem 10** *Let $G$ be a group acting on the function space $\mathcal{F}$ with type $\boldsymbol{\rho}$. Consider the manifold $\mathcal{Z}$ as the jet space of $\mathcal{F}$, and let $G$ act on $\mathcal{Z}$ via the prolongation of its action on $\mathcal{F}$. Suppose we have a moving frame $\alpha : \mathcal{Z} \to G$. Define $\boldsymbol{\mathcal{M}} : \mathbb{R}^2 \times \mathcal{F} \to \mathbb{R}^{c' \times c'}$ as*

$$\boldsymbol{\mathcal{M}}(\mathbf{x}, \mathbf{u}) = \left(\boldsymbol{\rho}'(\alpha(z))\right)^{-1}, \tag{14}$$

*where $z = (\mathbf{x}, \mathbf{u}(\mathbf{x}), \nabla \mathbf{u}(\mathbf{x}), \cdots, \nabla^d \mathbf{u}(\mathbf{x})) \in \mathcal{Z}$. Then $\boldsymbol{\mathcal{M}}$ is a type-$(\boldsymbol{\rho}, \boldsymbol{\rho}')$ equivariant matrix.*

So far, we have presented a complete framework and theoretical foundation for constructing equivariants. First, we compute the moving frame as outlined in Theorem 9. Next, we utilize Theorem 10 to derive an equivariant matrix from this moving frame. Finally, we combine the equivariant matrix with various invariants, as shown in Theorem 7, to generate a wide variety of equivariants. This approach provides a systematic way to build equivariant operators that can map features between any specified input and output types, thereby enabling a high degree of flexibility in the design of equivariant networks.

## 2.3 SupNorm normalized equivariants

In the previous subsection, we outline a theoretical method for constructing equivariants. Building upon this foundation, we can adopt more flexible strategies to obtain equivariants that are simpler in form and exhibit improved numerical stability in practice. A critical aspect of this process is the construction of equivariant matrices. Similar to the problem encountered with differential invariants, as discussed in (Li et al., 2024), direct computation of equivariants for the affine group may lead to complex expressions and potential division-by-zero issues. To mitigate these problems, we propose a new SupNorm normalization technique for the construction of more robust equivariant matrices.

While Theorem 7 shows that equivariant matrices can be used to construct equivariants, the relationship between equivariants and equivariant matrices is even more intricate: each column of an equivariant matrix is itself an equivariant. Conversely, if every column of a matrix-valued function is an independent equivariant, the entire function qualifies as an equivariant matrix. Thus, our objective is to ensure that each column of the matrix-valued function independently forms an equivariant.

Here, we consider the affine group $G = \{(s \cdot \tilde{\mathbf{A}}, \mathbf{b}) \mid s > 0, \tilde{\mathbf{A}} \in \mathbb{R}^{2 \times 2}, \det(\tilde{\mathbf{A}}) = 1, \mathbf{b} \in \mathbb{R}^2\}$, which is, more precisely, the connected component of the full affine group. The equi-affine

group, denoted as $\mathrm{SA}(2)$, is the subgroup of the affine group without scaling, defined as $\mathrm{SA}(2) = \{(\tilde{\mathbf{A}}, \mathbf{b}) \mid \tilde{\mathbf{A}} \in \mathbb{R}^{2 \times 2}, \det(\tilde{\mathbf{A}}) = 1, \mathbf{b} \in \mathbb{R}^2\}$. In (Li et al., 2024), SupNorm normalized differential invariants of the affine group are constructed by normalizing relative differential invariants which can be derived from the differential invariants of the subgroup $\mathrm{SA}(2)$. Inspired by this approach, we begin with the equivariant matrix for $\mathrm{SA}(2)$, and aim to transform each column into an equivariant of the affine group. To formalize this process, we begin with the definition of a *relative equivariant*:

**Definition 11** *Let $G$ be the affine group acting on the function space $\mathcal{F} = \{\mathbf{u} \mid \mathbf{u} : \mathbb{R}^2 \to \mathbb{R}^c\}$ with type $\boldsymbol{\rho}$. Let $s(g)$ denote the scaling factor of the affine group element $g$. We define $\bar{\mathcal{E}} : \mathbb{R}^2 \times \mathcal{F} \to \mathbb{R}^{c'}$ as a type-$(\boldsymbol{\rho}, \boldsymbol{\rho'})$ **relative equivariant** with power $m$ if it satisfies*

$$\bar{\mathcal{E}}(g \cdot \mathbf{x}, L_{\boldsymbol{\rho}}(g) \diamond \mathbf{u}) = \frac{1}{(s(g))^m} \cdot \boldsymbol{\rho'}(g) \cdot \bar{\mathcal{E}}(\mathbf{x}, \mathbf{u}), \quad \forall \mathbf{x} \in \mathbb{R}^2, \mathbf{u} \in \mathcal{F}, g \in G. \tag{15}$$

*If $\boldsymbol{\rho'} = \boldsymbol{\rho}_0$ (the trivial representation), $\bar{\mathcal{E}}$ is called a **relative invariant**.*

Then, we provide the following theorem to demonstrate our new SupNorm normalization technique for converting a relative equivariant into a full equivariant:

**Theorem 12** *Let $\mathcal{F}_k = \{\mathbf{v} \mid \mathbf{v} : \mathbb{R}^2 \to \mathbb{R}^k\}$ denote the set of smooth bounded functions on $\mathbb{R}^2$. Define the SupNorm on $\mathcal{F}_k$ as $\|\mathbf{v}\|_{\sup} \triangleq \sup_{\mathbf{x} \in \mathbb{R}^2} \|\mathbf{v}(\mathbf{x})\|_{\infty}$. Let $G$ be the affine group acting on the function space $\mathcal{F}_c$ with type $\boldsymbol{\rho}$. Suppose $\bar{\mathcal{E}} : \mathbb{R}^2 \times \mathcal{F}_c \to \mathbb{R}^{c'}$ is a type-$(\boldsymbol{\rho}, \boldsymbol{\rho'})$ relative equivariant with power $m_1$, and $\bar{\mathcal{I}} : \mathbb{R}^2 \times \mathcal{F}_c \to \mathbb{R}^k$ is a relative invariant with power $m_2 \neq 0$. Define $\mathcal{E} : \mathbb{R}^2 \times \mathcal{F}_c \to \mathbb{R}^{c'}$ as follows:*

$$\mathcal{E}(\mathbf{x}, \mathbf{u}) \triangleq \frac{1}{\left(\|\bar{\mathcal{I}}(\cdot, \mathbf{u})\|_{\sup}\right)^{\frac{m_1}{m_2}}} \cdot \bar{\mathcal{E}}(\mathbf{x}, \mathbf{u}). \tag{16}$$

*Then $\mathcal{E}$ is a type-$(\boldsymbol{\rho}, \boldsymbol{\rho'})$ equivariant.*

The above theorem provides a concrete method to transform relative equivariants into exact equivariants using SupNorm normalization. Next, we will demonstrate a typical example that applies Theorem 12 for clearer understanding and practical usage. For instance, we define $\boldsymbol{\rho}_0$ as the trivial representation, and $\boldsymbol{\rho}_1(g) = \mathbf{A}^{-\top}$, where $g = (\mathbf{A}, \mathbf{b}) \in G$. By computing the type-$(\boldsymbol{\rho}_0, \boldsymbol{\rho}_1)$ equivariant matrix for $\mathrm{SA}(2)$, we obtain

$$\begin{bmatrix} u_y u_{xx} - u_x u_{xy} & u_x \\ u_y u_{xy} - u_x u_{yy} & u_y \end{bmatrix}, \tag{17}$$

where $u_x, u_y, u_{xx}, u_{yy}$ and $u_{xy}$ denote the partial derivatives of the scalar function $u(x, y)$. Although this function is not directly an exact equivariant matrix for the affine group, each column is a type-$(\boldsymbol{\rho}_0, \boldsymbol{\rho}_1)$ relative equivariant. Specifically, the first column has a power of 2, and the second column has a power of 0, making it a true equivariant on its own. We use a power 4 relative invariant $u_{xx} u_{yy} - u_{xy}^2$ to apply SupNorm normalization to (17), yielding the following matrix:

$$\left[ \frac{1}{\mathcal{S}(u)} \begin{pmatrix} u_y u_{xx} - u_x u_{xy} \\ u_y u_{xy} - u_x u_{yy} \end{pmatrix}, \quad \begin{pmatrix} u_x \\ u_y \end{pmatrix} \right], \tag{18}$$

where $\mathcal{S}(u) = \|u_{xx} u_{yy} - u_{xy}^2\|_{\sup}^{\frac{1}{2}}$. This resulting matrix in (18) is a type-$(\boldsymbol{\rho}_0, \boldsymbol{\rho}_1)$ equivariant for the affine group. In contrast to directly computing affine equivariant matrices, which may involve complex fractional polynomial expressions prone to the division-by-zero issue, this method yields a numerically stable equivariant matrix. Such division-by-zero problems would only occur in the rare case where $u$ is a constant function, a scenario that is typically negligible in practical applications.

Furthermore, this method is also applicable to continuous subgroups of the affine group. For example, in the case of the rotation-scale group, one can compute the equivariant matrix for the rotation group to obtain relative equivariants, which can then be normalized to form an exact equivariant matrix using the same SupNorm normalization technique. For the scale group, one can start with the identity matrix and apply appropriate normalization to directly obtain the equivariant matrix.

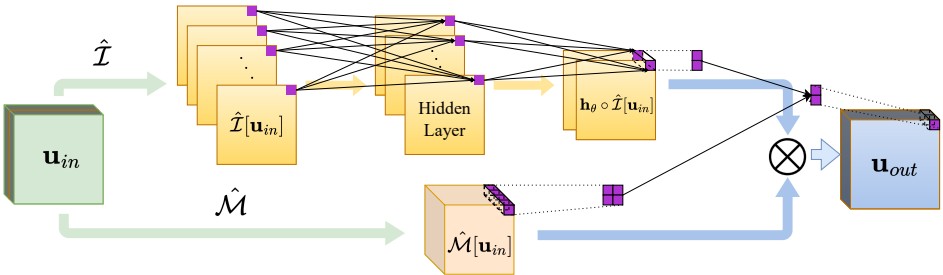

Figure 1: Architecture of the steerable EquivarLayer.

## 2.4 ARCHITECTURE OF EQUIVARLAYER

Building on the previous theoretical framework, we can derive the corresponding equivariants given the specified group representation types, leading to the construction of the desired equivariant operators. In practice, our ultimate goal is to design a parameterized steerable equivariant layer that can handle arbitrary specified input and output feature types.

Let the input and output group representations be denoted by $\boldsymbol{\rho}_{in}$ and $\boldsymbol{\rho}_{out}$, with dimensions $c$ and $c'$, respectively. If $\boldsymbol{\rho}_{out}$ is the trivial representation, we can adopt the structure of InvarLayer from (Li et al., 2024). The equivariant layer is defined as:

$$\mathbf{u}_{out} = \mathbf{h}_\theta \circ \hat{\boldsymbol{\mathcal{I}}}[\mathbf{u}_{in}], \tag{19}$$

where $\boldsymbol{\mathcal{I}}$ is a $k$-dimensional invariant formed by a set of basis invariants, $\hat{\boldsymbol{\mathcal{I}}}$ is the corresponding equivariant operator as defined in (4), and $\mathbf{h}_\theta$ is a learnable multi-layer perceptron (MLP) with input dimension $k$ and output dimension $c'$.

Next, we focus on the more general case where $\boldsymbol{\rho}_{out}$ is not a trivial representation, requiring equivariant operators defined by equivariants. In this scenario, the equivariant layer is defined as:

$$\mathbf{u}_{out} = \hat{\boldsymbol{\mathcal{M}}}[\mathbf{u}_{in}] \cdot \left( \mathbf{h}_\theta \circ \hat{\boldsymbol{\mathcal{I}}}[\mathbf{u}_{in}] \right), \tag{20}$$

where $\hat{\boldsymbol{\mathcal{I}}}$ and $\mathbf{h}_\theta$ follow the definitions in (19), $\boldsymbol{\mathcal{M}}$ is a type-$(\boldsymbol{\rho}_{in}, \boldsymbol{\rho}_{out})$ equivariant matrix, and $\hat{\boldsymbol{\mathcal{M}}}[\mathbf{u}] \triangleq \boldsymbol{\mathcal{M}}(\cdot, \mathbf{u})$. This formulation allows the layer to handle arbitrary input and output feature types. We refer to this general equivariant layer as a type-$(\boldsymbol{\rho}_{in}, \boldsymbol{\rho}_{out})$ *EquivarLayer* (see Figure 1). The layer defined in (19), where $\boldsymbol{\rho}_{out}$ is the trivial representation, can be viewed as a special case, specifically a type-$(\boldsymbol{\rho}_{in}, \boldsymbol{\rho}_0)$ EquivarLayer.

Moreover, we consider a common scenario where the input and output are multi-channel. Specifically, we focus on the setting where $\boldsymbol{\rho}_{in} = \oplus^{C_1} \boldsymbol{\rho}_a$ and $\boldsymbol{\rho}_{out} = \oplus^{C_2} \boldsymbol{\rho}_b$, meaning that both the input and output are direct sums of multiple channels of the group representations $\boldsymbol{\rho}_a$ and $\boldsymbol{\rho}_b$, respectively. While it is theoretically possible to handle this setting using the methods described earlier, we can leverage the structure of the group representations to simplify the computation. We define the multi-channel version of EquivarLayer (see Figure 2 in Appendix E) as follows:

$$\mathbf{u}_{out} = \begin{pmatrix} \hat{\boldsymbol{\mathcal{M}}}_\lambda[\mathbf{u}_{in}] \cdot \left( \mathbf{h}_\theta \circ \hat{\boldsymbol{\mathcal{I}}}[\mathbf{u}_{in}] \right)^{(1)} \\ \vdots \\ \hat{\boldsymbol{\mathcal{M}}}_\lambda[\mathbf{u}_{in}] \cdot \left( \mathbf{h}_\theta \circ \hat{\boldsymbol{\mathcal{I}}}[\mathbf{u}_{in}] \right)^{(C_2)} \end{pmatrix}, \tag{21}$$

where $\hat{\boldsymbol{\mathcal{M}}}_\lambda[\mathbf{u}_{in}] = \Sigma_{i=1}^{C_1} \lambda_i \cdot \hat{\boldsymbol{\mathcal{M}}}[\mathbf{u}_{in}^{(i)}]$. Here, $\lambda_i$ are learnable scalar parameters, $\boldsymbol{\mathcal{M}}$ is a type-$(\boldsymbol{\rho}_a, \boldsymbol{\rho}_b)$ equivariant matrix, $\mathbf{u}_{in}^{(i)}$ denotes the $i$-th channel vector function of $\mathbf{u}_{in}$, and $\left( \mathbf{h}_\theta \circ \hat{\boldsymbol{\mathcal{I}}}[\mathbf{u}_{in}] \right)^{(j)}$ represents the $j$-th channel vector function of $\mathbf{h}_\theta \circ \hat{\boldsymbol{\mathcal{I}}}[\mathbf{u}_{in}]$. This formulation enables efficient processing of multi-channel inputs and outputs while preserving the desired group equivariance.

## 3 STEERABLE EQUIVARLAYER FOR CANONICALIZATION

Canonicalization is a method for achieving invariance or equivariance without imposing constraints on the architecture of a neural network. By utilizing a canonicalization function, the input is mapped to a canonical sample before being passed to a non-equivariant prediction network, ensuring that the output remains invariant. In the case of equivariance, the input is mapped to its canonical form, the prediction is made, and the result is transformed back to its original position under the group action. Below, we give a formal definition of canonicalization in our context.

**Definition 13** *Let $G$ be a group acting on the function space $\mathcal{F}$ with type $\boldsymbol{\rho}$. A map $\varphi : \mathcal{F} \to G$ is called a **canonicalization function** if it satisfies*

$$\varphi(L_{\boldsymbol{\rho}}(g) \diamond \mathbf{u}) = \varphi(\mathbf{u}) \cdot g^{-1}, \quad \forall g \in G, \mathbf{u} \in \mathcal{F}. \tag{22}$$

To illustrate the process of canonicalization, we state the following theorem:

**Theorem 14** *Let $G$ be a group acting on function spaces $\mathcal{F}$ and $\mathcal{F}'$ with types $\boldsymbol{\rho}$ and $\boldsymbol{\rho}'$, respectively. Given a prediction operator $\psi_0 : \mathcal{F} \to \mathcal{F}'$, define $\psi : \mathcal{F} \to \mathcal{F}'$ as follows:*

$$\psi[\mathbf{u}] \triangleq L_{\boldsymbol{\rho}'}((\varphi(\mathbf{u}))^{-1}) \diamond \psi_0[L_{\boldsymbol{\rho}}(\varphi(\mathbf{u})) \diamond \mathbf{u}]. \tag{23}$$

*Then $\psi$ is equivariant, satisfying $\psi[L_{\boldsymbol{\rho}}(g) \diamond \mathbf{u}] = L_{\boldsymbol{\rho}'}(g) \diamond \psi[\mathbf{u}]$. If we remove $L_{\boldsymbol{\rho}'}((\varphi(\mathbf{u}))^{-1})$ in (23), the operator becomes invariant.*

The core component of canonicalization is the canonicalization function, which can be interpreted as a form of equivariant mapping. In the direct approach (Kaba et al., 2023), an equivariant network that produces a group element serves as the canonicalization function. Previous works have mostly focused on rotation groups for canonicalization, largely due to the absence of more general steerable equivariant networks. We aim to leverage the proposed steerable EquivarLayer to extend this concept to more general groups. Specifically, we focus on $G = \mathrm{GL}^+(2) \triangleq \{\mathbf{A} \mid \mathbf{A} \in \mathbb{R}^{2 \times 2}, \det(\mathbf{A}) > 0\}$. Thus, our goal is to construct a canonicalization function $\varphi : \mathcal{F} \to \mathbb{R}^{2 \times 2}$ that satisfies

$$\varphi(L_{\boldsymbol{\rho}}(\mathbf{A}) \diamond \mathbf{u}) = \varphi(\mathbf{u}) \cdot \mathbf{A}^{-1}, \quad \forall \mathbf{A} \in G, \mathbf{u} \in \mathcal{F}. \tag{24}$$

We proceed in two steps. Let $\boldsymbol{\rho}_c(\mathbf{A}) = \mathbf{A}^{-T}$, $\mathbf{A} \in G$. First, construct a map $\phi_1 : \mathcal{F} \to \tilde{\mathcal{F}}$, satisfying $\phi_1[L_{\boldsymbol{\rho}}(\mathbf{A}) \diamond \mathbf{u}] = L_{\boldsymbol{\rho}_c}(\mathbf{A}) \diamond \phi_1[\mathbf{u}]$, where $\tilde{\mathcal{F}} = \{\mathbf{v} \mid \mathbf{v} : \mathbb{R}^2 \to \mathbb{R}^{2 \times 2}\}$. Next, construct a second map $\phi_2 : \tilde{\mathcal{F}} \to \mathbb{R}^{2 \times 2}$ such that $\phi_2[L_{\boldsymbol{\rho}_c}(\mathbf{A}) \diamond \mathbf{v}] = \boldsymbol{\rho}_c(\mathbf{A}) \cdot \phi_2[\mathbf{v}]$. Composing these maps, $\phi = \phi_2 \circ \phi_1$ satisfies $\phi(L_{\boldsymbol{\rho}}(\mathbf{A}) \diamond \mathbf{u}) = \mathbf{A}^{-\top} \cdot \phi(\mathbf{u})$. Thus, $\phi^\top$ is a valid canonicalization function.

For the construction of $\phi_1$, note that the function space $\tilde{\mathcal{F}} = \{\mathbf{v} \mid \mathbf{v} : \mathbb{R}^2 \to \mathbb{R}^{2 \times 2}\}$ with type $\boldsymbol{\rho}_c$ can be associated with the vector-valued function space $\mathcal{F}' = \{\mathbf{u} \mid \mathbf{u} : \mathbb{R}^2 \to \mathbb{R}^4\}$, corresponding to the group representation $\boldsymbol{\rho}_c \oplus \boldsymbol{\rho}_c$. Therefore, we can employ a type-$(\boldsymbol{\rho}, \boldsymbol{\rho}_c \oplus \boldsymbol{\rho}_c)$ EquivarLayer to serve as $\phi_1$. The output of this EquivarLayer is then reshaped to match the required form in $\tilde{\mathcal{F}}$. In practice, this equivariant network can consist of multiple layers, as long as the input feature type of the first layer is type $\boldsymbol{\rho}$ and the output feature type of the final layer is type $\boldsymbol{\rho}_c \oplus \boldsymbol{\rho}_c$.

For the construction of $\phi_2$, we need to select one matrix from a matrix-valued function while maintaining the equivariance condition. For this purpose, we propose a module called *Det-Pooling*, which selects the matrix with the largest absolute determinant from the matrix-valued function. The key idea is that under the transformation $L_{\boldsymbol{\rho}_c}(\mathbf{A})$, each point is mapped to a new location and its associated matrix are also transformed, but the matrix with the largest absolute determinant remains the largest one. The following theorem defines $\phi_2$ and demonstrates its property:

**Theorem 15** *Assume $\mathbf{v} \in \tilde{\mathcal{F}}$ and that $|\det(\mathbf{v}(\mathbf{x}))|$ has a unique maximum over $\mathbf{x} \in \mathbb{R}^2$. Let **Det-Pooling** $\phi_2 : \tilde{\mathcal{F}} \to \mathbb{R}^{2 \times 2}$ be defined as*

$$\phi_2(\mathbf{v}) \triangleq \mathbf{v} \left( \underset{\mathbf{x} \in \mathbb{R}^2}{\mathrm{argmax}} \, |\det(\mathbf{v}(\mathbf{x}))| \right). \tag{25}$$

*Then $\phi_2$ satisfies the equivariance property*

$$\phi_2[L_{\boldsymbol{\rho}_c}(\mathbf{A}) \diamond \mathbf{v}] = \boldsymbol{\rho}_c(\mathbf{A}) \cdot \phi_2[\mathbf{v}], \quad \forall \mathbf{A} \in G. \tag{26}$$

Det-Pooling effectively tracks the matrix with the largest absolute determinant, thus ensuring equivariance under group transformations. In addition, it reduces the likelihood of selecting a singular matrix, as this would only occur if $\mathbf{v}(\mathbf{x})$ is singular at every point, which is a rare scenario in practice.

In summary, we can construct $\phi_1$ using steerable EquivarLayers and employ Det-Pooling as $\phi_2$, together forming the canonicalization function. Notably, this method can also be extended to handle subgroups of $\mathrm{GL}^+(2)$, making it a versatile approach for broader applications.

## 4 EXPERIMENT

To evaluate the effectiveness of the proposed steerable EquivarLayer, we apply it to canonicalization in combination with an unconstrained prediction network. Our experiments focus on image classification tasks using the MNIST dataset and its transformed versions under different group actions. Specifically, we assess the performance of our method on three non-compact continuous groups: the $\mathrm{GL}^+(2)$ group, the rotation-scale group, and the scale group.

### 4.1 EXPERIMENT SETUP

We use a ResNet50 model pre-trained on the ImageNet-1K dataset as the prediction network for our experiments, which has 23.5M parameters. The model is fine-tuned on the MNIST dataset using three distinct strategies: (1) Vanilla: Training on the original MNIST dataset without any data augmentation. (2) Mild augmentation: Training with data augmentation, involving mild affine transformations, including rotation within $\pm 10$ degrees, scaling between $0.9 - 1.1$, and shear within $\pm 5$ degrees. (3) Full augmentation: Training with the same data augmentation used in the transformed test set. These three strategies represent common training practices, and the resulting models serve as our baselines for comparison.

For the canonicalization function, we integrate EquivarLayer with a ResNet32 architecture, where the convolutional layers are replaced with type-$(\boldsymbol{\rho}_0, \boldsymbol{\rho}_0)$ EquivarLayers, and the output module is replaced with a type-$(\boldsymbol{\rho}_0, \boldsymbol{\rho}_c \oplus \boldsymbol{\rho}_c)$ EquivarLayer followed by Det-Pooling as described in Section 3. In each experiment, each model uses EquivarLayers tailored to the corresponding group action and contains fewer than 0.4M parameters. For convenience, we call the whole model EquivarLayer. Inspired by (Mondal et al., 2023), we adopt a training stage for alignment. Although the canonicalization function can map elements within a group orbit to a canonical representative, we aim to ensure that this representative is aligned with the orientation the prediction network is familiar with, such as the upright position. EquivarLayer is trained independently of the prediction network by generating random transformation matrices applied to the training images. The loss is defined as the mean squared error (MSE) between the identity matrix and the product of EquivarLayer's output and the corresponding transformation matrix. During inference, the trained EquivarLayer is connected to a pre-trained prediction network that uses mild augmentation. The EquivarLayer canonicalizes the input images (see Figure 3 in Appendix E), which are then fed into the prediction network for classification. More experiment details are given in Appendix C.

### 4.2 EVALUAION

Typically, full data augmentation helps improve a model's performance on transformed datasets, but it often comes at the cost of reduced performance on the original dataset. Mild augmentation, on the other hand, may enhance performance on the original dataset, but it usually fails to achieve low test error on transformed datasets. Our objective is to leverage canonicalization to improve performance on transformed datasets while maintaining competitive performance on the original dataset.

### 4.2.1 $\mathrm{GL}^+(2)$ GROUP

Table 1: Test error (%) on MNIST and MNIST-$\mathrm{GL}^+(2)$.

| Method | MNIST | MNIST-$\mathrm{GL}^+(2)$ |
|---|---|---|
| Vanilla | $0.87_{\pm 0.07}$ | $54.22_{\pm 0.98}$ |
| Mild Augmentation | $\mathbf{0.45}_{\pm 0.03}$ | $47.28_{\pm 1.51}$ |
| Full Augmentation | $1.19_{\pm 0.04}$ | $1.52_{\pm 0.08}$ |
| EquivarLayer Canonicalizer (ours) | $0.81_{\pm 0.08}$ | $\mathbf{1.32}_{\pm 0.08}$ |

We conduct experiments on the original MNIST dataset and its transformed version, MNIST-$GL^+(2)$, which undergoes random transformations from $GL^+(2)$ group including arbitrary rotations between $-180$ to $180$ degrees, scaling between $0.8 - 1.2$, and shear within $\pm 10$ degrees. Each experiment is repeated five times with independent random seeds, and we report the mean $\pm$ standard deviation of test error in Table 1.

Mild augmentation improves performance on both MNIST and MNIST-$GL^+(2)$ compared to the vanilla model, though the results on MNIST-$GL^+(2)$ remain relatively sub-optimal. Full augmentation achieves good performance on MNIST-$GL^+(2)$ but performs the worst on the original MNIST dataset. Our method, combining mild augmentation with canonicalization, outperforms full augmentation on MNIST-$GL^+(2)$ while maintaining competitive performance on the original MNIST dataset, second only to mild augmentation.

#### 4.2.2 ROTATION-SCALE GROUP

Table 2: Test error (%) on MNIST and MNIST-RS.

| Method | MNIST | MNIST-RS |
|---|---|---|
| Vanilla | $0.87_{\pm 0.07}$ | $54.18_{\pm 0.93}$ |
| Mild Augmentation | $\mathbf{0.45}_{\pm 0.03}$ | $47.36_{\pm 1.39}$ |
| Full Augmentation | $1.19_{\pm 0.07}$ | $1.65_{\pm 0.06}$ |
| EquivarLayer Canonicalizer (ours) | $0.71_{\pm 0.08}$ | $\mathbf{1.06}_{\pm 0.07}$ |

We also evaluate our method on the rotation-scale group, where MNIST-RS includes arbitrary rotations between $-180$ to $180$ degrees and scaling between $0.8 - 1.2$. The results, averaged over five independent runs, are shown in Table 2. Our method yields the best performance on MNIST-RS while maintaining low test error on MNIST, close to that of mild augmentation.

#### 4.2.3 SCALE GROUP

Table 3: Test error (%) on MNIST and MNIST-Scale.

| Method | MNIST | MNIST-Scale |
|---|---|---|
| Vanilla | $0.87_{\pm 0.07}$ | $8.72_{\pm 2.18}$ |
| Mild Augmentation | $0.45_{\pm 0.03}$ | $1.62_{\pm 0.30}$ |
| Full Augmentation | $0.49_{\pm 0.04}$ | $0.67_{\pm 0.04}$ |
| EquivarLayer Canonicalizer (ours) | $\mathbf{0.44}_{\pm 0.03}$ | $\mathbf{0.64}_{\pm 0.06}$ |

We further evaluate our method on the scale group, where the MNIST-Scale dataset involves random scaling transformations with factors ranging from $0.8$ to $1.6$. The results are presented in Table 3. Since the scale transformation is relatively simple, mild augmentation already yields a significant improvement on MNIST-Scale, and full augmentation achieves near-perfect test error on both the MNIST and MNIST-Scale datasets. However, our method not only achieves the best performance on the MNIST-Scale dataset but also slightly outperforms all baselines on the original MNIST dataset.

## 5 CONCLUSION

In this paper, we introduce the steerable EquivarLayer, extending the InvarLayer framework to support equivariance with arbitrary input and output feature types for the affine group and its continuous subgroups. This marks the first time steerability is achieved in networks for the affine group. We also develop the novel Det-Pooling module, which enables the steerable EquivarLayer's applicability to canonicalization, thus extending the method to handle more complex matrix groups. Our experiments demonstrate the effectiveness of this approach, showing that the steerable EquivarLayer, when used as a canonicalization function, outperforms traditional data augmentation methods.

Our framework holds great potential for future research, particularly in extending beyond the 2D plane, as the theoretical foundations apply to higher-dimensional spaces. Exploring equivariance on more general manifolds and applying steerable EquivarLayers to a wider variety of groups is another promising direction. Additionally, expanding the application of this method to a broader range of tasks, from computer vision to scientific problems, presents valuable avenues for further study.

ACKNOWLEDGMENT

Z. Lin was supported by National Key R&D Program of China (2022ZD0160300) and the NSF China (No. 62276004).

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

# A RELATED WORKS

## A.1 EQUIVARIANT NETWORKS

Equivariant networks have been widely applied in various fields, such as computer vision (Esteves et al., 2019; Weiler et al., 2018; Klee et al., 2023; Xu et al., 2023; 2024), sciences (Liao & Smidt, 2022; Wang et al., 2024; Park et al., 2024b), reinforcement learning and robotics (Wang et al., 2022; Park et al., 2022; 2024a; Hu et al., 2024; Huang et al., 2022; 2023; 2024). One primary approach to building group equivariant networks is based on G-CNNs (Cohen & Welling, 2016a) where feature maps are considered as functions defined on a group and the group action on the feature is essentially a permutation over the domain. From this perspective, various equivariant operators are designed for cycle or dihedral groups (Cohen & Welling, 2016a; He et al., 2021; Romero et al., 2020; Worrall & Brostow, 2018; Shen et al., 2020; Cohen et al., 2019). Besides discrete groups, Cohen et al. (2018) have extended such an approach to handle inherent $SO(3)$ equivariance for data defined on the sphere. Finzi et al. (2020) further proposed LieConv that is equivariant to transformations from common Lie groups. However, it fails to handle more complex Lie groups like the affine group due to the difficulty of sampling the Haar measure. Furthermore, MacDonald et al. (2022) overcome this challenge by designing special sampling strategies on the affine group. However, since the domain of feature is defined on groups, it is unavoidable to sample when dealing with the affine group, which would aggravate the computation burden as the increasing of samples and layers. Mironenco & Forré (2024) tackles this problem by decomposing a large group into smaller ones and sampling them to enhance sample efficiency. It requires sampling from the $GL(n)$-invariant measure of positive definite matrices. Instead, they approximate this using a log-normal distribution, which may theoretically introduce imperfect equivariance.

Another approach to constructing equivariant networks originates from steerable CNNs (Cohen & Welling, 2016b; Weiler & Cesa, 2019). In this framework, the feature maps are viewed as fields and are transformed according to a specific group representation under the action of group transformation. Actually, the feature maps in the first approach precisely correspond to fields with regular representation. This approach can directly handle continuous groups since the group transformation can be preserved by the linear transformation of the vectors on the fields. Based on this method, works (Cohen & Welling, 2016b; Jenner & Weiler, 2021; He et al., 2022) have developed operators to handle rotations on the 2D plane. Weiler et al. (2018); Fuchs et al. (2020); Shen et al. (2022) further design equivariant layers to handle transformations in 3D space like $E(3)$ and its subgroups. Zhdanov et al. (2023) provide a general framework for building neural networks equivariant to translations and subgroups of $O(n)$. However, it is crucial to recognize that these methods are mainly designed to tackle rigid transformations and are not well-suited for more general affine transformations.

Besides these two mainstream approaches, some works construct equivariant networks based on differential invariants (Sangalli et al., 2022; 2023; Li et al., 2024). In particular, Li et al. (2024) propose a novel equivariant layer, InvarLayer, which achieves affine equivariance without the need for sampling from groups. However, the input and output representation types of InvarLayer are constrained to the trivial representation. Our work is a further extension of InvarLayer, capable of handling any given representation type of the affine group.

## A.2 EQUIVARIANCE BY CANONICALIZATION FUNCTIONS

In equivariant networks, equivariance is typically achieved by imposing equivariance constraints on each layer. Recently, Kaba et al. (2023) propose a novel method to incorporate equivariance that no longer requires layer-wise architectural constraints. Their approach introduces canonicalization functions that learn mappings transforming inputs into canonicalized samples. In this framework, only these canonicalization functions need to satisfy certain equivariance constraints to incorporate additional symmetries, thereby explicitly decoupling the equivariant and non-equivariant parts of the networks. Mondal et al. (2023) further utilize canonicalization functions to enable the adaptation of large pre-trained models to be equivariant.

# B   DETAILED PROOFS

## B.1   PROOF OF PROPOSITION 3

*Proof.* $\forall \mathbf{u} \in \mathcal{F}, g \in G, \mathbf{x} \in \mathbb{R}^2$, we have

$$
\begin{aligned}
\hat{\mathcal{I}}[L_{\boldsymbol{\rho}}(g) \diamond \mathbf{u}](\mathbf{x}) &= \mathcal{I}(\mathbf{x}, L_{\boldsymbol{\rho}}(g) \diamond \mathbf{u}) \\
&= \mathcal{I}(g^{-1} \cdot \mathbf{x}, \mathbf{u}) \\
&= \hat{\mathcal{I}}[\mathbf{u}](g^{-1} \cdot \mathbf{x}) \\
&= (L_{\boldsymbol{\rho}_0}(g) \diamond \hat{\mathcal{I}}[\mathbf{u}])(\mathbf{x}).
\end{aligned}
\tag{27}
$$

Thus, $\hat{\mathcal{I}}$ satisfies the condition for equivariance, completing the proof. □

## B.2   PROOF OF PROPOSITION 5

*Proof.* $\forall \mathbf{u} \in \mathcal{F}, g \in G, \mathbf{x} \in \mathbb{R}^2$, we have

$$
\begin{aligned}
\hat{\mathcal{E}}[L_{\boldsymbol{\rho}}(g) \diamond \mathbf{u}](\mathbf{x}) &= \mathcal{E}(\mathbf{x}, L_{\boldsymbol{\rho}}(g) \diamond \mathbf{u}) \\
&= \boldsymbol{\rho}'(g) \cdot \mathcal{E}(g^{-1} \cdot \mathbf{x}, \mathbf{u}) \\
&= \boldsymbol{\rho}'(g) \cdot \hat{\mathcal{E}}[\mathbf{u}](g^{-1} \cdot \mathbf{x}) \\
&= (L_{\boldsymbol{\rho}'}(g) \diamond \hat{\mathcal{E}}[\mathbf{u}])(\mathbf{x}).
\end{aligned}
\tag{28}
$$

Hence, $\hat{\mathcal{E}}[L_{\boldsymbol{\rho}}(g) \diamond \mathbf{u}] = L_{\boldsymbol{\rho}'}(g) \diamond \hat{\mathcal{E}}[\mathbf{u}]$. □

## B.3   PROOF OF THEOREM 7

*Proof.* $\forall \mathbf{u} \in \mathcal{F}, g \in G, \mathbf{x} \in \mathbb{R}^2$, we have

$$
\begin{aligned}
\mathcal{E}(g \cdot \mathbf{x}, L_{\boldsymbol{\rho}}(g) \diamond \mathbf{u}) &= \boldsymbol{\mathcal{M}}(g \cdot \mathbf{x}, L_{\boldsymbol{\rho}}(g) \diamond \mathbf{u}) \cdot \mathcal{I}(g \cdot \mathbf{x}, L_{\boldsymbol{\rho}}(g) \diamond \mathbf{u}) \\
&= \boldsymbol{\rho}'(g) \cdot \boldsymbol{\mathcal{M}}(\mathbf{x}, \mathbf{u}) \cdot \mathcal{I}(\mathbf{x}, \mathbf{u}) \\
&= \boldsymbol{\rho}'(g) \cdot \mathcal{E}(\mathbf{x}, \mathbf{u})
\end{aligned}
\tag{29}
$$

This proves that $\mathcal{E}$ is a type-$(\boldsymbol{\rho}, \boldsymbol{\rho}')$ equivariant. □

## B.4   PROOF OF THEOREM 10

*Proof.* By the definition of the group action on $\mathcal{Z}$, we have

$$
g \cdot z = (g \cdot \mathbf{x}, \mathbf{u}_g(g \cdot \mathbf{x}), \nabla \mathbf{u}_g(g \cdot \mathbf{x}), \cdots, \nabla^d \mathbf{u}_g(g \cdot \mathbf{x})),
\tag{30}
$$

where $\mathbf{u}_g = L_{\boldsymbol{\rho}}(g) \diamond \mathbf{u}$. Then we will show that $\boldsymbol{\mathcal{M}}(g \cdot \mathbf{x}, L_{\boldsymbol{\rho}}(g) \diamond \mathbf{u}) = \boldsymbol{\rho}'(g) \cdot \boldsymbol{\mathcal{M}}(\mathbf{x}, \mathbf{u})$.

$$
\begin{aligned}
\boldsymbol{\mathcal{M}}(g \cdot \mathbf{x}, L_{\boldsymbol{\rho}}(g) \diamond \mathbf{u}) &= (\boldsymbol{\rho}'(\alpha(g \cdot z)))^{-1} \\
&= \left(\boldsymbol{\rho}'(\alpha(z) \cdot g^{-1})\right)^{-1} \\
&= \left(\boldsymbol{\rho}'(\alpha(z)) \cdot \boldsymbol{\rho}'(g^{-1})\right)^{-1} \\
&= \boldsymbol{\rho}'(g) \cdot (\boldsymbol{\rho}'(\alpha(z)))^{-1} \\
&= \boldsymbol{\rho}'(g) \cdot \boldsymbol{\mathcal{M}}(\mathbf{x}, \mathbf{u}).
\end{aligned}
\tag{31}
$$

Thus, $\boldsymbol{\mathcal{M}}$ is a type-$(\boldsymbol{\rho}, \boldsymbol{\rho}')$ equivariant matrix. □

### B.5 PROOF OF THEOREM 12

*Proof.* First, we compute the SupNorm under the group action:

$$
\begin{aligned}
\|\bar{\mathcal{I}}(\cdot, L_{\boldsymbol{\rho}}(g) \diamond \mathbf{u})\|_{\mathrm{sup}} &= \sup_{\mathbf{x} \in \mathbb{R}^2} \bar{\mathcal{I}}(\mathbf{x}, L_{\boldsymbol{\rho}}(g) \diamond \mathbf{u}) \\
&= \sup_{\mathbf{x} \in \mathbb{R}^2} \bar{\mathcal{I}}(g \cdot \mathbf{x}, L_{\boldsymbol{\rho}}(g) \diamond \mathbf{u}) \\
&= \sup_{\mathbf{x} \in \mathbb{R}^2} \left( \frac{1}{(s(g))^{m_2}} \bar{\mathcal{I}}(\mathbf{x}, \mathbf{u}) \right) \\
&= \frac{1}{(s(g))^{m_2}} \cdot \sup_{\mathbf{x} \in \mathbb{R}^2} \bar{\mathcal{I}}(\mathbf{x}, \mathbf{u}) \\
&= \frac{1}{(s(g))^{m_2}} \cdot \|\bar{\mathcal{I}}(\cdot, \mathbf{u})\|_{\mathrm{sup}}.
\end{aligned}
\tag{32}
$$

Next, we apply this property to $\mathcal{E}$ defined in (16):

$$
\begin{aligned}
\mathcal{E}(g \cdot \mathbf{x}, L_{\boldsymbol{\rho}}(g) \diamond \mathbf{u}) &= \frac{1}{\left( \|\bar{\mathcal{I}}(\cdot, L_{\boldsymbol{\rho}}(g) \diamond \mathbf{u})\|_{\mathrm{sup}} \right)^{\frac{m_1}{m_2}}} \cdot \bar{\mathcal{E}}(g \cdot \mathbf{x}, L_{\boldsymbol{\rho}}(g) \diamond \mathbf{u}) \\
&= \frac{1}{\left( \frac{1}{(s(g))^{m_2}} \|\bar{\mathcal{I}}(\cdot, \mathbf{u})\|_{\mathrm{sup}} \right)^{\frac{m_1}{m_2}}} \cdot \frac{1}{(s(g))^{m_1}} \cdot \boldsymbol{\rho}'(g) \cdot \bar{\mathcal{E}}(\mathbf{x}, \mathbf{u}) \\
&= \frac{1}{\left( \|\bar{\mathcal{I}}(\cdot, \mathbf{u})\|_{\mathrm{sup}} \right)^{\frac{m_1}{m_2}}} \cdot \boldsymbol{\rho}'(g) \cdot \bar{\mathcal{E}}(\mathbf{x}, \mathbf{u}) \\
&= \boldsymbol{\rho}'(g) \cdot \mathcal{E}(\mathbf{x}, \mathbf{u}).
\end{aligned}
\tag{33}
$$

Thus, $\mathcal{E}$ satisfies the definition of a type-$(\boldsymbol{\rho}, \boldsymbol{\rho}')$ equivariant. $\qquad\square$

### B.6 PROOF OF THEOREM 14

*Proof.* We focus on proving the equivariant case. Denote $L_{\boldsymbol{\rho}}(g) \diamond \mathbf{u} = \mathbf{u}_g$. $\forall g \in G, \mathbf{u} \in \mathcal{F}$, we have

$$
\begin{aligned}
\psi[L_{\boldsymbol{\rho}}(g) \diamond \mathbf{u}] &= L_{\boldsymbol{\rho}'}((\varphi(\mathbf{u}_g))^{-1}) \diamond \psi_0[L_{\boldsymbol{\rho}}(\varphi(\mathbf{u}_g)) \diamond \mathbf{u}_g] \\
&= L_{\boldsymbol{\rho}'}(g \cdot (\varphi(\mathbf{u}))^{-1}) \diamond \psi_0[L_{\boldsymbol{\rho}}(\varphi(\mathbf{u}) \cdot g^{-1}) \diamond \mathbf{u}_g] \\
&= L_{\boldsymbol{\rho}'}(g) \diamond L_{\boldsymbol{\rho}'}((\varphi(\mathbf{u}))^{-1}) \diamond \psi_0[L_{\boldsymbol{\rho}}(\varphi(\mathbf{u})) \diamond L_{\boldsymbol{\rho}}(g^{-1}) \diamond L_{\boldsymbol{\rho}}(g) \diamond \mathbf{u}] \\
&= L_{\boldsymbol{\rho}'}(g) \diamond L_{\boldsymbol{\rho}'}((\varphi(\mathbf{u}))^{-1}) \diamond \psi_0[L_{\boldsymbol{\rho}}(\varphi(\mathbf{u})) \diamond \mathbf{u}] \\
&= L_{\boldsymbol{\rho}'}(g) \diamond \psi[\mathbf{u}].
\end{aligned}
\tag{34}
$$

Thus, $\psi$ satisfies the equivariance condition. $\qquad\square$

### B.7 PROOF OF THEOREM 15

*Proof.* Denote $\mathbf{v}_{\mathbf{A}} \triangleq L_{\boldsymbol{\rho}_c}(\mathbf{A}) \diamond \mathbf{v}$, i.e., $\mathbf{v}_{\mathbf{A}}(\mathbf{x}) = \mathbf{A}^{-\top} \cdot \mathbf{v}(\mathbf{A}^{-1}\mathbf{x})$.
Let $\mathbf{x}^{\star} = \mathrm{argmax}_{\mathbf{x} \in \mathbb{R}^2} |\det(\mathbf{v}(\mathbf{x}))|$. We first show that $\mathbf{A}\mathbf{x}^{\star} = \mathrm{argmax}_{\mathbf{x} \in \mathbb{R}^2} |\det(\mathbf{v}_{\mathbf{A}}(\mathbf{x}))|$.
$\forall \mathbf{A} \in G$ and $\mathbf{x} \neq \mathbf{A}\mathbf{x}^{\star} \in \mathbb{R}^2$, we have $\mathbf{A}^{-1}\mathbf{x} \neq \mathbf{x}^{\star}$. Then

$$
\begin{aligned}
|\det(\mathbf{v}_{\mathbf{A}}(\mathbf{x}))| &= |\det(\mathbf{A}^{-\top} \cdot \mathbf{v}(\mathbf{A}^{-1}\mathbf{x}))| \\
&= |\det(\mathbf{A}^{-\top})| \cdot |\det(\mathbf{v}(\mathbf{A}^{-1}\mathbf{x}))| \\
&< |\det(\mathbf{A}^{-\top})| \cdot |\det(\mathbf{v}(\mathbf{x}^{\star}))| \\
&= |\det(\mathbf{A}^{-\top} \cdot \mathbf{v}(\mathbf{A}^{-1}\mathbf{A}\mathbf{x}^{\star}))| \\
&= |\det(\mathbf{v}_{\mathbf{A}}(\mathbf{A}\mathbf{x}^{\star}))|.
\end{aligned}
\tag{35}
$$

Thus, $\mathbf{A}\mathbf{x}^\star = \mathrm{argmax}_{\mathbf{x}\in\mathbb{R}^2} |\det(\mathbf{v_A}(\mathbf{x}))|$. Now, for the equivariance of $\phi_2$, we have:

$$\begin{aligned}
\phi_2[L_{\boldsymbol{\rho}_c}(\mathbf{A}) \diamond \mathbf{v}] &= \phi_2[\mathbf{v_A}] \\
&= \mathbf{v_A}(\mathbf{A}\mathbf{x}^\star) \\
&= \mathbf{A}^{-\top} \cdot \mathbf{v}(\mathbf{A}^{-1}\mathbf{A}\mathbf{x}^\star) \\
&= \mathbf{A}^{-\top} \cdot \mathbf{v}(\mathbf{x}^\star) \\
&= \boldsymbol{\rho}_c(\mathbf{A}) \cdot \phi_2[\mathbf{v}].
\end{aligned} \tag{36}$$

Thus, the proof is complete. □

## C  EXPERIMENTAL DETAILS

**Experiment Configuration.** In all experiments in Section 4, each training dataset consists of 50,000 images, and each test dataset consists of 10,000 images. All experiments are conducted on an NVIDIA RTX 3090 GPU.

**Prediction Networks.** We utilize a ResNet50 model pre-trained on the ImageNet-1k dataset as the prediction network for our experiments. The model is fine-tuned using SGD with a learning rate of $10^{-3}$, decay of $5 \times 10^{-4}$, and momentum of $0.9$ for a duration of 50 epochs. The learning rate scheduler reduces the learning rate at one-third and one-half of the total epochs, multiplying it by a factor of $0.1$ at each milestone. The batch size for datasets is set to 128.

**Canonicalization Networks.** We incorporate EquivarLayers into a ResNet32 architecture, replacing the convolutional layers with type-$(\boldsymbol{\rho}_0, \boldsymbol{\rho}_0)$ EquivarLayers. Additionally, the output module is substituted with a type-$(\boldsymbol{\rho}_0, \boldsymbol{\rho}_c \oplus \boldsymbol{\rho}_c)$ EquivarLayer, followed by Det-Pooling, as described in Section 3. During the training stage for alignment, the canonicalization network is trained by applying randomly generated transformation matrices, corresponding to the evaluation group, to the training images from the MNIST dataset. The model is trained using an AdamW optimizer with a learning rate of $2 \times 10^{-3}$ and employs a cosine annealing scheduler for 200 epochs. The batch size for the datasets is set to 128.

## D  ADDITIONAL EXPERIMENTS

### D.1  MEASUREMENT OF COMPUTATIONAL COMPLEXITY

Table 4: The memory usage, FLOPs, and inference time per image.

| Input Size | Memory (MB) | FLOPs | Time (ms) |
|---|---|---|---|
| $32 \times 32$ | 0.03 | $6.86 \times 10^4$ | 1.7 |
| $64 \times 64$ | 0.12 | $2.74 \times 10^5$ | 1.8 |
| $128 \times 128$ | 0.50 | $1.10 \times 10^6$ | 1.8 |
| $256 \times 256$ | 2.00 | $4.39 \times 10^6$ | 1.8 |
| $512 \times 512$ | 8.00 | $1.76 \times 10^7$ | 2.8 |
| $1024 \times 1024$ | 32.00 | $7.02 \times 10^7$ | 8.7 |
| $2048 \times 2048$ | 128.00 | $2.81 \times 10^8$ | 32.4 |

We measure the time and space complexity of a type-$(\boldsymbol{\rho}_0, \boldsymbol{\rho}_c \oplus \boldsymbol{\rho}_c)$ EquivarLayer used in Section 4.2.1 and present the numerical results in Table 4. Following (Li et al., 2024), we use torchstat to calculate the memory usage and the FLOPs required for model inference, and perform explicit timing measurements during inference. The results of memory usage and FLOPs indicate that the time and space complexity of the model grows linearly with the input image size. For large input sizes, the inference time increases proportionally, while for smaller input sizes, it grows more slowly. This behavior is likely due to the underlying parallelism and optimizations in low-level computations, as noted in (Li et al., 2024).

### D.2  ADDITIONAL DATASETS

As shown in Tables 5, 6, and 7, we include additional experiments on Fashion-MNIST (Xiao et al., 2017) and its transformed variants. Specifically, we construct three datasets, Fashion-GL$^+$(2),

Fashion-RS, and Fashion-Scale, by zero-padding original Fashion-MNIST images to $40 \times 40$ and applying corresponding group transformations. The transformation parameters are consistent with those detailed in Section 4. Besides, we employ the same prediction networks and canonicalization networks as in Section 4, including ResNet50 with different data augmentation strategies and EquivarLayers for three groups. During inference, EquivarLayer is connected to a pre-trained prediction network that uses mild augmentation. ResNet50 is fine-tuned on Fashion-MNIST using SGD with a learning rate of $10^{-3}$, weight decay of $5 \times 10^{-4}$, momentum of 0.9, and a batch size of 128 for 50 epochs. The learning rate scheduler reduces the rate by a factor of 0.1 at one-third and one-half of the total epochs. The canonicalization network is trained for alignment using an AdamW optimizer with a learning rate of $2 \times 10^{-3}$ and a cosine annealing scheduler for 400 epochs. The batch size for datasets is set to 128. The experiments are repeated five times and we report mean $\pm$ std of the test error.

Tables 5 and 6 demonstrate that while full augmentation enhances a model's performance on transformed datasets, it often sacrifices performance on the original dataset; mild augmentation, on the other hand, may improve performance on the original dataset but generally underperforms on transformed datasets. By combining prediction networks trained with mild augmentation and our EquivarLayer canonicalizers, we achieve improved performance on transformed datasets while maintaining competitive results on the original dataset. Compared to full augmentation, this approach delivers better performance on the original dataset and achieves comparable results on transformed datasets. In Table 7, probably due to the simplicity of scale transformations, full augmentation also performs well on the original dataset. Nevertheless, we can still observe that canonicalization enhances the performance on transformed datasets for models trained with mild augmentation. It is worth noting that the performance of this approach is closely tied to the capability of the prediction network. Combining a powerful pre-trained model with the canonicalizer is a promising strategy, especially since off-the-shelf pre-trained models typically do not undergo full augmentation.

Table 5: Test error (%) on Fashion and Fashion-GL$^{+}(2)$.

| Method | Fashion | Fashion-GL$^{+}(2)$ |
|---|---|---|
| Vanilla | $6.58_{\pm 0.19}$ | $71.83_{\pm 1.07}$ |
| Mild Augmentation | $5.46_{\pm 0.08}$ | $66.35_{\pm 2.01}$ |
| Full Augmentation | $7.62_{\pm 0.16}$ | $8.38_{\pm 0.10}$ |
| EquivarLayer Canonicalizer (ours) | $6.71_{\pm 0.17}$ | $8.60_{\pm 0.27}$ |

Table 6: Test error (%) on Fashion and Fashion-RS.

| Method | Fashion | Fashion-RS |
|---|---|---|
| Vanilla | $6.58_{\pm 0.19}$ | $71.69_{\pm 0.79}$ |
| Mild Augmentation | $5.46_{\pm 0.08}$ | $65.82_{\pm 1.60}$ |
| Full Augmentation | $7.70_{\pm 0.15}$ | $8.28_{\pm 0.04}$ |
| EquivarLayer Canonicalizer (ours) | $6.41_{\pm 0.17}$ | $8.34_{\pm 0.28}$ |

Table 7: Test error (%) on Fashion and Fashion-Scale.

| Method | Fashion | Fashion-Scale |
|---|---|---|
| Vanilla | $6.58_{\pm 0.19}$ | $18.09_{\pm 1.06}$ |
| Mild Augmentation | $5.46_{\pm 0.08}$ | $10.26_{\pm 0.42}$ |
| Full Augmentation | $5.36_{\pm 0.05}$ | $5.73_{\pm 0.24}$ |
| EquivarLayer Canonicalizer (ours) | $5.46_{\pm 0.05}$ | $6.84_{\pm 0.18}$ |

### D.3 ADDITIONAL BASELINES

To provide a more comprehensive evaluation, we conduct additional experiments with equivariant networks of the corresponding groups as baselines, including InvarLayers (Li et al., 2024) for the three groups and the scale-equivariant model SESN (Sosnovik et al., 2019), involving various data augmentation strategies. All these baseline models adopt the ResNet50 architecture and are trained on the MNIST dataset for 50 epochs using an AdamW optimizer with a learning rate of $1 \times 10^{-3}$ and a cosine annealing scheduler. Furthermore, we combine EquivarLayers as canonicalization networks with equivariant baseline models trained using mild augmentation as prediction networks. We repeat

each experiment five times and report the mean $\pm$ std of the test error in Tables 8, 9, and 10 (Aug = Augmentation, MA = Mild Augmentation, FA = Full Augmentation, EquivarCan = EquivarLayer Canonicalizer).

Regarding comparisons with equivariant network baselines, we emphasize an important distinction: while EquivarLayer and other equivariant network baselines are all equivariant models, they serve different roles in the canonicalization experiments. These approaches are not mutually exclusive; rather, they are complementary and can be effectively combined. These equivariant networks, when used as prediction networks, outperform ResNet50 under the same augmentation settings. Thus, we integrate EquivarLayers as canonicalizers with equivariant networks as prediction networks, achieving further improved performance.

As shown in Tables 8, 9, and 10, equivariant baseline models with mild augmentation, despite their theoretical equivariance, may exhibit suboptimal performance on transformed datasets. A possible reason is that the large number of layers and the presence of several downsampling operations may weaken the equivariance in practice. However, when combined with canonicalization, these models maintain competitive performance on the original dataset while significantly enhancing their performance on transformed datasets, achieving results comparable to those of full augmentation.

Table 8: Test error (%) on MNIST and MNIST-GL$^+$(2).

| Model | MNIST | MNIST-GL$^+$(2) |
|---|---|---|
| InvarLayer w/o Aug | $0.72_{\pm0.05}$ | $41.38_{\pm1.28}$ |
| InvarLayer w/ MA | $0.47_{\pm0.02}$ | $36.71_{\pm0.63}$ |
| InvarLayer w/ FA | $0.58_{\pm0.03}$ | $0.87_{\pm0.05}$ |
| EquivarCan + InvarLayer (ours) | $0.63_{\pm0.05}$ | $0.92_{\pm0.02}$ |

Table 9: Test error (%) on MNIST and MNIST-RS.

| Model | MNIST | MNIST-RS |
|---|---|---|
| InvarLayer w/o Aug | $0.57_{\pm0.04}$ | $41.36_{\pm1.30}$ |
| InvarLayer w/ MA | $0.43_{\pm0.03}$ | $37.31_{\pm0.97}$ |
| InvarLayer w/ FA | $0.57_{\pm0.04}$ | $0.75_{\pm0.05}$ |
| EquivarCan + InvarLayer (ours) | $0.57_{\pm0.03}$ | $0.74_{\pm0.04}$ |

Table 10: Test error (%) on MNIST and MNIST-Scale.

| Model | MNIST | MNIST-Scale |
|---|---|---|
| SESN w/o Aug | $0.43_{\pm0.04}$ | $3.99_{\pm0.92}$ |
| SESN w/ MA | $0.34_{\pm0.04}$ | $1.03_{\pm0.20}$ |
| SESN w/ FA | $0.35_{\pm0.03}$ | $0.44_{\pm0.03}$ |
| InvarLayer w/o Aug | $0.43_{\pm0.02}$ | $4.54_{\pm0.70}$ |
| InvarLayer w/ MA | $0.37_{\pm0.03}$ | $1.53_{\pm0.19}$ |
| InvarLayer w/ FA | $0.35_{\pm0.02}$ | $0.38_{\pm0.03}$ |
| EquivarCan + SESN (ours) | $0.33_{\pm0.05}$ | $0.43_{\pm0.04}$ |
| EquivarCan + InvarLayer (ours) | $0.36_{\pm0.02}$ | $0.40_{\pm0.02}$ |

### D.4 EQUIVARLAYERS OF DIFFERENT TYPES

Previously, we focused on the application of EquivarLayer to canonicalization, specifically utilizing a type-$(\boldsymbol{\rho}_0, \boldsymbol{\rho}_c \oplus \boldsymbol{\rho}_c)$ EquivarLayer. Our proposed framework is general and supports EquivarLayers with arbitrary input and output feature types. To validate EquivarLayers of various types, one straightforward approach is to specify the feature types of the network's hidden layers, analogous to steerable CNNs in (Weiler & Cesa, 2019). For simplicity, we refer to scalar features as type-0 and vector features as type-1. We implement several three-layer affine equivariant networks as image classifiers, where both the input and output consist of type-0 features, and the hidden layers can involve either type-0 or type-1 features. These models include EquivarLayers that map from type-0 to type-0, type-0 to type-1, type-1 to type-0, and type-1 to type-1. We train these networks on MNIST and evaluate them on both MNIST and MNIST-GL$^+$(2). The results are summarized in Table 11. Models with different types exhibit varying performance, highlighting the flexibility and expanded design space that EquivarLayer provides for network architectures.

Table 11: Test error (%) on MNIST and MNIST-GL$^+$(2).

| Model | Type | MNIST | MNIST-GL$^+$(2) |
|---|---|---|---|
| InvarLayer | 0000 | $0.96_{\pm 0.07}$ | $8.07_{\pm 0.28}$ |
| EquivarLayer | 0100 | $1.06_{\pm 0.09}$ | $6.56_{\pm 0.73}$ |
| EquivarLayer | 0010 | $1.21_{\pm 0.05}$ | $6.51_{\pm 0.42}$ |
| EquivarLayer | 0110 | $1.44_{\pm 0.06}$ | $6.43_{\pm 0.22}$ |

### D.5 EQUIVARLAYERS FOR DYNAMICAL SYSTEM PREDICTION

To further demonstrate the increased generality of our framework, we conduct numerical experiments on the prediction of dynamical systems. The goal is to use our equivariant network model to predict the evolution of a dynamical system governed by a partial differential equation (PDE). Specifically, we consider the PDE: $\partial_t \mathbf{u} = \nabla_{\mathbf{x}} \mathbf{u} \cdot \mathbf{u}$, where $\mathbf{u}(t, \mathbf{x}) = (u_1(t, \mathbf{x}), u_2(t, \mathbf{x}))^\top$, and $\mathbf{x} = (x_1, x_2)^\top$. This PDE describes a vector field that exhibits affine symmetry. If the initial vector field of the dynamical system undergoes an affine transformation, the vector field at each subsequent time step is transformed in the same manner. Following (Wang et al., 2020), we predict the vector field at the next time step based on the past $l$ steps of the vector field. Here we set $l$ to 3.

For the dataset, we randomly generate functions with specific conditions as the initial values for the PDE and use the Runge-Kutta method to iteratively compute the evolution of the dynamical system. We retain sequences by sampling every 5th step of iteration and employ a rolling window approach to generate subsequences of length 4, where 3 steps are used as input to predict the next step and 1 step serves as the ground truth output. From each sequence, we extract 48 such subsequences. Each input data point has the shape (3, 2, 64, 64), while the output has the shape (1, 2, 64, 64). We generate 2310 sequences with different initial conditions and split the dataset into a 5:2 ratio, using 1650 sequences for training (corresponding to 79,200 data points) and 660 sequences for testing (31,680 data points).

We use a three-layer affine equivariant EquivarLayer, where the input, output, and hidden layers consist of vector features. As a baseline, we employ an SO(2) equivariant steerable CNN (Weiler & Cesa, 2019), which also uses vector features in each layer, and has approximately the same number of parameters as EquivarLayer. The models are evaluated using the Root Mean Square Error (RMSE) between the forward predictions and the ground truth over all pixels. As a reference for absolute values, if the output is simply set to 0, the RMSE is $5231.071 \times 10^{-5}$. As shown in Table 12, EquivarLayer achieves significantly higher precision, demonstrating its superior performance.

Table 12: RMSE ($10^{-5}$) on Test Set.

| Model | RMSE | Number of Parameters |
|---|---|---|
| SO(2) Steerable CNN | $6.881_{\pm 0.150}$ | 107 K |
| EquivarLayer (ours) | $0.218_{\pm 0.012}$ | 104 K |

## E ADDITIONAL FIGURES

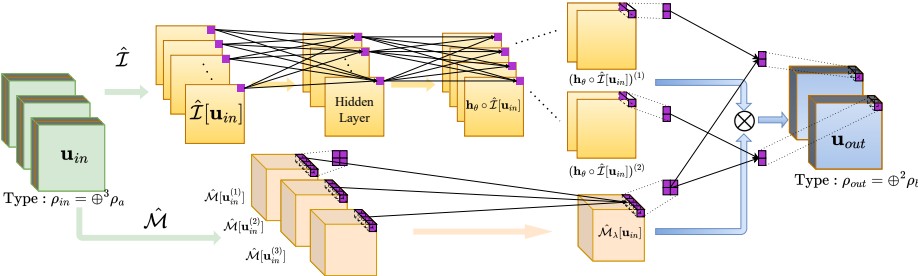

Figure 2: Multi-channel version of the steerable EquivarLayer.

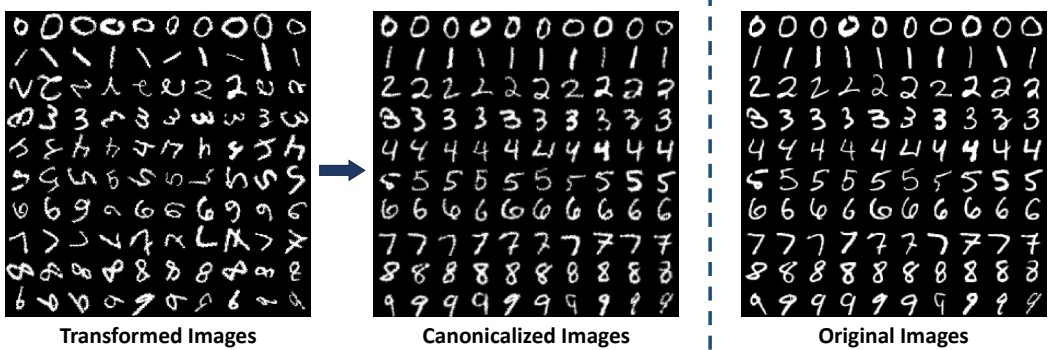

Figure 3: Visualization of the canonicalization process.

