# OpenReview forum: "Affine Steerable Equivariant Layer for Canonicalization of Neural Networks"
_ICLR.cc/2025/Conference — ICLR 2025 Poster_

### Official Review · Reviewer_5XxQ · 2024-11-03

**Soundness:** 3
**Presentation:** 2
**Contribution:** 3
**Rating:** 6
**Confidence:** 4

**Summary:**

In summary, this paper contributes the following:
- An extension of the equivariant steerable layers for the 2D affine group proposed in [1] from type-0 (scalar-type) feature maps (in the terminology of [2, 3]) to other types utilizing equivariant moving frames [4], specifically by linear combination of the equivariants produced by frames using type-0 features as weights,
- An adaptation of the above layer for invariant or equivariant canonicalization [5] of non-equivariant backbones for the 2D affine group, using global max-pooling of 2-channel type-1 feature with respect to an invariant quantity (absolute determinant),
- Experiments on invariant canonicalization of a pre-trained ResNet50 backbone for MNIST image classification under affine, rotation-scaling, and scaling transformations.

[1] Li et al. Affine equivariant networks based on differential invariants (2024)

[2] Cohen and Welling, Steerable CNNs (2017)

[3] Thomas et al. Tensor field networks: Rotation- and translation-equivariant neural networks for 3D point clouds (2018)

[4] Olver, Modern developments in the theory and applications of moving frames (2015)

[5] Kaba et al. Equivariance with learned canonicalization functions (2022)

**Strengths:**

- S1. The use of equivariant moving frames to extend type-0 features to other types is technically sound, and original as far as I am aware in the context of steerable networks. (In a broader context, the authors may find [6] and its application to Lorentz canonicalization [7] related, as they also linearly combine equivariants using learned invariant features.)
- S2. The considered application for canonicalization using global invariant pooling is technically sound as far as I can tell, and properly shows the utility of the proposed layer since canonicalization [5] only using type-0 features is not straightforward.
- S3. The experiments show that the approach is able to perform canonicalization of a non-equivariant backbone for invariant MNIST classification under affine transformations, which has been unavailable in current literature before as far as I am aware.

[5] Kaba et al. Equivariance with learned canonicalization functions (2022)

[6] Villar et al. Scalars are universal: Equivariant machine learning, structured like classical physics (2021)

[7] Nguyen et al. Learning symmetrization for equivariance with orbit distance minimization (2023)

**Weaknesses:**

- W1. A proper comparison against fully equivariant networks for classification (e.g., InvarPDEs-Net and InvarLayer from [1]) is not included in the experiments. Including these baselines would strengthen the paper, especially since canonicalization may benefit from pre-training in non-invariant domains (e.g., ImageNet-1k in this paper) unlike fully equivariant networks.
- W2. It seems possible to use the moving frame in Definition 8 (or the equivariant matrix in Theorem 10) directly for canonicalization without multiplication by type-0 features (Eq. (10)), as this coincides with the definition of equivariant frames in frame averaging [8]. Adding this as a baseline would strengthen the results, as it can be understood as ablating trainability from canonicalization.
- W3. It was unclear to me how the definition and property of moving frames given in Definition 8 and Theorem 9 logically leads to the construction of equivariants using relative equivariants given in Section 2.3.
- W4. The paper demonstrates a single application of the proposed layer as the last layer of a canonicalization network, which extracts 2-channel type-1 features from type-0 features, while all other layers of the network map between type-0 features [1]. This misses (1) feature maps of type >1, (2) layers taking non-scalar features as input, and (3) layers mapping between non-scalar features, all of which are possible within the proposed framework. Testing (a subset) of these would strengthen the paper. A straightforward way is to modify InvarPDEs-Net and InvarLayer from [1] to have non-scalar hidden features, analogous to steerable networks in [2].

Minor typo
- Line 13: InvarLayer -> InvarLayer (Li et al. 2024)

[1] Li et al. Affine equivariant networks based on differential invariants (2024)

[2] Cohen and Welling, Steerable CNNs (2017)

[8] Puny et al. Frame averaging for invariant and equivariant network design (2021)

**Questions:**

- Q1. Is it necessary that equivariant matrices are square matrices, as implied in Definition 6? As far as I understand, it should be that their number of columns can be flexible, while the number of rows is specified by the representation \rho'.

---

> ### Comment · Reviewer_5XxQ · 2024-11-27
>
> Thank you for the response and supplementing the experimental results. I have some follow-up questions after reading the response and other reviews.
>
> - On R1, Tables 2 and 3 show that InvarLayer fails catastrophically on MNIST-RS and MNIST-GL+(2). Are there specific reasons in their failure? To my understanding, InvarLayer and EquivarLayer-canonicalized ResNet-50 are both invariant to the symmetry groups of the interest by construction, and I do not see clear reasons why InvarLayer should show such high error rates on MNIST-RS and MNIST-GL+(2), compared to their low error rates on MNIST.
>
> - On R2 for Reviewer 8iML, I have reservations about the statement "... It is comparable to key advancements in the domain, such as the progression from G-CNNs [8] to steerable CNNs [5-7]..." A key contribution of [5] is proposing to incorporate steerability to equivariant CNNs for the first time. This work contributes an application of the idea of [5] to affine groups, while also building upon another prior work (Li et al. InvarLayer). Comparing the contributions as written in the response seems misleading.
>
> Other than the above, I would like to hear the opinions of other reviewers, especially Reviewer uxCw on the raised concerns, before making further decisions.
>
> [5] Cohen and Welling, Steerable CNNs (2017)

---

### Official Review · Reviewer_sBuw · 2024-11-04

**Soundness:** 3
**Presentation:** 2
**Contribution:** 3
**Rating:** 6
**Confidence:** 3

**Summary:**

This paper extends the InvarLayer (Li et al., 2024), which uses differential invariants to achieve affine equivariance, by proposing the EquivarLayer. EquivarLayer combines a novel equivariant matrix with differential invariants to generalize from invariance to equivariance. The paper also introduces Det-pooling, which enables EquivarLayer to achieve canonicalization. This canonicalization capability is assessed by using the EquivarLayer model to canonicalize transformed MNIST images, followed by classification with a ResNet-50. The results show improved performance compared to ResNet-50 with data augmentation.

**Strengths:**

1. This paper introduces a novel and efficient method for extending the affine-invariant layer to an affine steerable equivariant layer. Achieving this in deep learning is challenging, as the affine group has six degrees of freedom, and sampling from this group often incurs prohibitively high computational costs.
2. The proposed ResNet-32 model contains only 0.4 million parameters, which is impressive. This compact model size may enable the exploration of very deep equivariant networks.

**Weaknesses:**

1. Experiment Design: Using invariant classification to demonstrate the effectiveness of the proposed EquivarLayer for canonicalization is somewhat indirect. The true strengths or limitations of EquivarLayer are obscured by the downstream ResNet-50 prediction network, yet five pages are dedicated to deriving the EquivarLayer. Since canonicalization is a downstream task, its accuracy depends significantly on the model’s ability to preserve information—a quality typically evaluated by learning equivariant features, performing invariant classification, or computing equivariant error. Key questions remain, such as whether EquivarLayer improves upon InvarLayer (as it is a generalization) and how it compares to other steerable methods achieving subgroups of affine group equivariance.
2. Experiment Setup and Comparability: The experiment setup limits comparability with other methods. The train/test split is not provided, and the scale factor range (0.8 to 1.2 or 1.6) differs from that of other scale-equivariant papers, which often use a range of [0.3, 1]. Additionally, key scale-equivariant works, such as [1], are not cited. These omissions make it challenging to compare this model with existing equivariant CNNs, especially since no equivariant network baselines are included in the experimental section.
3. Clarity in Derivations: The derivations following Eq. (16) in Section 2.3 are unclear. The notation for symbols like $u_x, u_y$ is not adequately defined, and it was only by referring to Li et al. (2024) that I understood they represent gradients. While this is a minor issue, a more significant concern is the lack of discussion around $\alpha$ from Eq. (14). Instead, the paper provides an example of an equivariant matrix (Eq. 17) without explaining the derivation of $\alpha$.
Additionally, one of the advantages of equivariance is its ability to preserve information. While the proposed method is computationally efficient, it may sacrifice some information, as it limits gradients to second order and relies on differential invariants for local features. It remains unclear if the gradient order relates to information preservation. The derivations also do not clarify why only first- and second-order gradients are used or how the equivariant matrix is derived.

[1] Sosnovik, Ivan, Michał Szmaja, and Arnold Smeulders. "Scale-Equivariant Steerable Networks." International Conference on Learning Representations.

**Questions:**

1. Can this approach be extended to handle reflections, specifically for cases where det(A) < 0?

---

### Official Review · Reviewer_uxCw · 2024-11-04

**Soundness:** 2
**Presentation:** 3
**Contribution:** 2
**Rating:** 6
**Confidence:** 3

**Summary:**

The paper proposes a framework for constructing equivariant layers based on the method of moving frames; under this framework it is shown how the class of input/output group representations that can be employed is larger than previously shown. The authors adapt the jet-space based construction to the case where we are working with an induced representation also acting on the codomain of the signals. The framework is then employed in the context of a canonicalization framework, where the proposed equivariant layer with respect to the affine group is used for the canonicalization function. A primary focus of this work is the affine group, more specifically the affine group of the plane and its subgroups that go beyond euclidean transformations (rotation-scale, $\textnormal{GL}(2, \mathbb{R})$). The methodology is evaluated on a set of image classification tasks employing transformed MNIST datasets, where it is contrasted with a standard data augmentation approach. The experimental validation focuses on the delta in performance on transformed/non-transformed test sets.

**Strengths:**

- The paper is generally clear in its presentation, with each mathematical object being well-introduced and described. The authors present a framework which indeed treats a larger class of group representations than seen for this framework.
- While there has been some recent work in this area, the problem of constructing equivariant/invariant layers with respect to larger transformation groups is generally under-explored and potentially useful. In this sense, I am also happy to see work employing the method of moving frames and the use of differential invariants, which may yield different complexity constraints and/or scaling behaviour on the spectrum of methods imposing inductive biases.
- Intertwining canonicalization and traditional methods for constructing equivariant layers seems like a useful avenue to explore.

**Weaknesses:**

I will categorize the issues I find with the paper into three categories: (1) Claimed novelty; (2) Lack of presentation of concrete examples given the level of generality claimed and (3) experimental validation.
I want to note from the onset that I am open to raising my score, and the degree to which I am penalizing (lack of) novelty here is very much based on what the authors themselves claim.

- (1) The paper uses language that supports the notion where a lot of the concepts are novel (in the sense of new mathematical objects) rather than, their application in this form in the context of machine learning being novel.
- Considering the framework presented up to section 2.3, I would like to highlight obviously the work of Olver himself in [4] where the same framework of constructing equivariant maps using moving frames based on his previous work is summarized (ending up with the abstract operator of eq. (13)). Directly related is the work of Sangalli et. al which also uses the prolongation of the group action on the jet space for invariantization (based on Olver's work) in the context of image and volumetric data [1-3]. From [1] appendix B, the general framework is directly applicable taking X = $\mathbb{R}^2$, $Y = \mathbb{R}^{c}$, $Z = \mathbb{R}^{c^{'}}$, we arrive at the same principle where an equivariant operator $\mathcal{F} \to \mathcal{F}^{'}$ on the function spaces is defined implicitly once a G-invariant operator on the product space $X \times Y$ is available ($g \cdot (x, u) = (gx, \rho(g)u) = (gx, u)$ ($\rho = \rho_0$), corresponding to $g \cdot f(x) = f(g^{-1}x)$). In this work the notation is less clear and it appears that $Y = \mathcal{F}$ indicating a higher order operator, however the differential invariants are always (correctly) constructed with the same $x \in X$ argument i.e. $\hat{\mathcal{I}}(f)(x) = I(x, f(x))$ (we make use of differential invariants dependent on $x$). It should be highlighted then that the 'equivariant' $\mathcal{E}: X \times Y \to Z$ presented here is a $G$-equivariant map on the product space respecting $\mathcal{E} \circ \rho(g) = \rho^{'}(g) \circ \mathcal{E}$ , and would correspond to the case where $\rho \neq \rho_0$ ((16) in [1]). This is indeed a different group action and the authors seek to effectively work with induced representations which also acts on the codomain $L_{\rho}(g)(f)(x) = \rho(g)f(g^{-1}x)$, however it should not be presented as a departure from this framework if one is simply changing the representation used.
- The authors should also try and clarify further how their framework differs from [6] in its usage of the sup-normalized differential invariants. Are the polynomial relative invariants of [6] and the relative invariants here the same? I would also highlight [5] and [7] which also deal with differential invariants of the affine group of the plane, where the zero division errors are treated with a different formulation (see [5] (34) and (35)).
- Regarding (2), I would have liked to see more concrete examples of the framework being employed in the context where the additional flexibility in terms of input/output representations is valuable e.g. beyond image data, see for example [9]. I am not concerned with SOTA results, simply a validation that the increased generality allows the use of the differential invariants for a larger set of applications and modalities. It would also be useful for practitioners if more concrete examples of the realized (beyond theoretical) operators is presented.
- Regarding (3), it would be useful to quantify more clearly the relationship between the amount of data augmentation used and the size of the dataset for each experiment. More importantly, what would be very useful here is to have a thorough presentation of the time/space complexity of each layer employing this framework, see e.g. [1] appendix D or [6], with the inclusion of standard (3-channel) image data.

[1] "Moving frame net: SE(3)-equivariant network for volumes", Sangalli et. al, 2023
[2] "Differential invariants for SE(2)-equivariant networks", Sangalli et. al, 2023
[3] "Equivariant Deep Learning Based on Scale-Spaces and Moving Frames", Sangalli 2023
[4] "Using Moving Frames to Construct Equivariant Maps", Peter Olvar (March 2024)
[5] "Affine Differential Invariants for Invariant Feature Point Detection", Tuznik et. al 2018
[6] "Affine Equivariant Networks Based on Differential Invariants", Li et. al, 2024
[7] "Affine invariant detection: edge maps, anisotropic diffusion, and active contours", Olver et.al 1999
[8] "On Relative Invariants", Fels & Olver 1997
[9] "Generalizing Convolutional Neural Networks for Equivariance to Lie Groups on Arbitrary Continuous Data" Finzi et. al 2020

**Questions:**

Beyond the issues highlighted before:
- In the case where the output representation acting on the codomain is not operating on scalar fields, do we have any restrictions with respect to the class of activation functions that can be used as it is the case for steerable methods which operate on vector fields? Do we have to use norm non-linearities?
- In the appendix it is stated "Mironenco & Forre (2024) tackles this problem by decomposing a large group into smaller ones and sampling them to enhance sample efficiency. However, it requires sampling on certain measures that may be impractical, such as GL(n)-invariant measure of positive definite matrices.". I'm not sure I understand what limitation is described here. As far as I understand, sampling from this measure is indeed possible (see e.g. [10]) and the SPD manifold has quite extensive applications.
- I don't understand how the Det-Pooling ensures a parametrization on the group $\textnormal{GL}(2, \mathbb{R})$. It seems we are mapping to the space of $\mathbb{R}^{2 \times 2}$ matrices and it is simply assumed that this is a group element?
- Is there a connection between this work and Olver's work on relative invariants [8]?

[10] Riemannian Gaussian Distributions on the Space of Symmetric Positive Definite Matrices, Said et. al 2015.

---

> ### Comment · Reviewer_uxCw · 2024-12-02
>
> The authors have made a strong effort in responding to the weaknesses mentioned and I've raised my score. To explain my reasoning for not increasing my score further (hopefully the authors can find some useful feedback here): I think the main weaknesses of the current paper revolve around being able to showcase/translate the increased generalization that comes with being able to use different $(\rho, \rho’)$-representations (point (2) in the original review) into concrete operators that can be applied to different modalities. Non-scalar field representations (which the theoretical framework of the paper unlocks) are more commonly employed outside of imaging data. The dynamical system experiment is appreciated, and a good step in this direction. As the authors state "We anticipate that researchers from various scientific domains could explore and apply our framework to problems with similar properties", and I think showcasing how operators within this framework can be implemented concretely for various domains and what their limitations are compared to 'traditional' steerable methods (not based on moving frames/invariants) would greatly improve the paper (since one could see the realization of 'larger class of theoretical operators' -> 'concrete implementations for different modalities' -> limitations, as in e.g. [1]).
>
> > For an EquivarLayer whose output is a vector field, applying activation functions while preserving equivariance is subject to certain conditions. However, it is important to emphasize that the EquivarLayer itself is inherently nonlinear, which distinguishes it significantly from steerable CNNs [6,7] or PDO-based methods [8-10]. This intrinsic nonlinearity eliminates the necessity for additional activation functions between layers.
>
> I'm not sure I agree that the nonlinearity baked into the layer removes the need for (other) nonlinearities. In my view this is an empirical claim that would need to be shown, since in general different activation functions can influence training dynamics and generalization/performance significantly. To be clear, I don't have a problem if this is a limitation of the framework in this case where it is applied using canonicalization, it would simply be better to know as others/future work can address it. This is in some sense an example of what I stated before, e.g. limitations that might appear in some settings and should be clarified.
>
> > The key lies in whether the resulting matrix is invertible. By definition, Det-Pooling selects the matrix with the largest absolute determinant from the matrix-valued function $v(x)$. While this does not guarantee invertibility in a strict mathematical sense, it maximizes the likelihood of the output matrix being invertible. The output will be a singular matrix if and only if $v(x)$ is singular for all x, which is an rare occurrence in practice.
>
> To make sure my understanding is correct. What is the shape of the tensor produced by  $\phi_{1}: \mathcal{F} \to \mathcal{\tilde{F}}$ in practice, and what do the axes represent? Am I understanding correctly that $\phi_{1}$ is deterministic, i.e. not learned? Are we pooling on what would be the channel axis?
>
> [1] A Practical Method for Constructing Equivariant Multilayer Perceptrons for Arbitrary Matrix Groups, Finzi 2021

---

> ### Comment · Reviewer_uxCw · 2024-12-03
>
> Thank you for the additional clarifications and comments.
>
> > In our experiments, the shape of the tensor produced by $\phi_1$ is $(2 \times 2, 28, 28)$, where the 4 channels correspond to the 4 elements of a $2 \times 2$ matrix, and $(28,28)$ represent the spatial axes. $\phi_1$ is a learnable equivariant model. We perform Det-Pooling on the spatial axes according to the absolute value of the determinant at each spatial point, which is analogous to conventional global pooling. The input with shape $(2 \times 2, 28, 28)$ results in an output of a $2 \times 2$ matrix (or equivalently, a 4-dimensional vector in the implementation).
>
> The authors might already be aware of this, but as a final future work suggestion, the map $\phi_{1}$ could probably be turned into one which produces an element of $\textnormal{GL}(2, \mathbb{R})$ directly since $\mathbb{R}^{2 \times 2}$ is isomorphic to the Lie algebra $\mathfrak{g}$ of $\textnormal{GL}(2, \mathbb{R})$ once a basis is chosen and from $\mathfrak{g}$ we can map to $\textnormal{GL}(2, \mathbb{R})$ using the group or Riemannian exponential (with the later being surjective).

---

### Official Review · Reviewer_8iML · 2024-11-04

**Soundness:** 3
**Presentation:** 2
**Contribution:** 2
**Rating:** 5
**Confidence:** 3

**Summary:**

In this work, the authors propose a steerable and affine group equivariant layer, EquivarLayer. They introduce a novel pooling technique that can facilitate canonicalization. The empirical experiments presented were conducted on MNIST and MNIST scale datasets.

**Strengths:**

1. This paper presents a novel pooling layer that keeps equivariance to affine groups intact.

**Weaknesses:**

1. The paper conducts experiments solely on MNIST datasets and skips existing methods like [1] for comparison.

2. The contribution does not seem significant and the experiments section is limited.

3. The motivation for using the proposed approach given the already existing methods that conduct canonicalization, and steerable layers, is unclear to me after going over the paper.

4. Definition 4, Proposition 5 as well as Theorem 7 can be improved in terms of formalizing as well as readability of equations.

[1] Implicit Convolutional Kernels for Steerable CNNs, Zhdanov et al.

**Questions:**

1. Could you present results with additional datasets along with a comparison with existing methods? For example, above mentioned Steerable CNNs as well as ablation with image datasets like CIFAR10/CIFAR100?


2. The addition of simple canonization as presented in Kaba et al compared to the proposed method would be helpful.


3. How does the proposed method perform to roto-translation and scale?


4. How does this method scale with data, dimensions of invariants as well as group size?

---

### Meta-Review · Area_Chair_qFqC · 2024-12-19

**Metareview:**

The paper proposes the “EquivarLayer,” extending the InvarLayer to support affine-equivariant transformations with steerability. The EquivarLayer is built upon the framework of moving frames and differential invariants, aiming to handle more general transformations within the affine group and its subgroups. As a key application, they integrate the EquivarLayer with a Det-Pooling module to achieve canonicalization. The results are demonstrated in MNIST dataset and its transformed variants and showed that EquivarLayer-based canonicalization can improve the performance of a non-equivariant pre-trained model.

The paper initially received somewhat mixed reviews. The major concerns raised by the reviewers include (1) weak technical contribution, since the work is largely based on prior methods on invariant layers for affine groups (InvarLayer), moving frames, and differentiable invariants, (2) limited evaluation, since the experiments are conducted only on MNIST (and its variants) while the prior works explored more diverse datasets, (3) limited applications, since the work demonstrates its effectiveness only on canonicalization while the capability of equivariant layers itself was not clearly presented. In response, the authors provided a comprehensive rebuttal, including results on more diverse datasets (e.g., Fashion MNIST, results on classification without canonicalization, and clarification of the contributions over prior works.  After the author's feedback, the reviewers found the additional evidence compelling, generally agreeing that while more rigorous experiments and further clarity would be welcome, the presented contributions are meaningful.

AC carefully read the paper, reviews, and discussions, and concurs with the reviewers' decision. The authors should incorporate the additional clarifications and experiment results presented in the rebuttal to the final version of the paper.

**Additional Comments On Reviewer Discussion:**

Reviewer 8iML initially questioned the novelty and practical significance of the contribution, arguing that experiments on MNIST were too limited and comparisons with existing steerable layers were insufficient. Reviewer uxCw called for more clarity regarding the claimed novelty, the relationship to previous moving frame methods, and the lack of examples illustrating non-scalar input-output representations. Reviewer sBuw focused on the limited evaluation using only MNIST, requesting more diverse datasets and the inclusion of stronger baselines, along with careful augmentation strategies to facilitate fair comparisons. Reviewer 5XxQ suggested comparing the proposed approach with fully equivariant models and showcasing more complex representations. In response, the authors provided additional results and clarifications, which successfully resolved many of these concerns.

---

### Decision · Program_Chairs · 2025-01-22

Accept (Poster)